# Effects of Inorganic Salts on the Heterogeneous OH Oxidation of Organic Compounds: Insights from Methylglutaric Acid-Ammonium Sulfate

Hoi Ki Lam[1], Sze Man Shum[1], James F. Davies[2], Mijung Song[3], Andreas Zuend[4], and Man Nin Chan[1,5,*]

[1]Earth System Science Programme, Faculty of Science, The Chinese University of Hong Kong, Hong Kong, China

[2]Department of Chemistry, University of California Riverside, Riverside, CA, USA

[3]Department of Earth and Environmental Sciences, Chonbuk National University, Jeollabuk-do, Republic of Korea

[4]Department of Atmospheric and Oceanic Sciences, McGill University, Montreal, Québec, Canada

[5]The Institute of Environment, Energy, and Sustainability, The Chinese University of Hong Kong, Hong Kong, China

Corresponding author: mnchan@cuhk.edu.hk

## Abstract

Atmospheric particles, consisting of inorganic salts, organic compounds and a varying amount of water, can continuously undergo heterogeneous oxidation initiated by gas-phase oxidants at the particle surface, changing the composition and properties of particles over time. To date, most studies focus on the chemical evolution of pure organic particles upon oxidation. To gain more fundamental insights into the effects of inorganic salts on the heterogeneous kinetics and chemistry of organic compounds, we investigate the heterogeneous OH oxidation of 3-methylglutaric acid (3-MGA) particles and particles containing both 3-MGA and ammonium sulfate (AS) in an organic-to-inorganic mass ratio of 2 in an aerosol flow tube reactor at a high relative humidity of 85.0 %. The molecular information of the particles before and after OH oxidation is obtained using the Direct Analysis in Real Time (DART), a soft atmospheric pressure ionization source, coupled to a

high-resolution mass spectrometer. Optical microscopy measurements reveal that 3-MGA-AS particles are in a single liquid phase prior to oxidation at high relative humidity. Particle mass spectra show that $C_6$ hydroxyl and $C_6$ ketone functionalization products are the major products formed upon OH oxidation in the absence and presence of AS, suggesting that the dissolved salt does not

significantly affect reaction pathways. The dominance of $C_6$ hydroxyl products over $C_6$ ketone products could be explained by the intermolecular hydrogen abstraction by tertiary alkoxy radicals formed at the methyl-substituted tertiary carbon site. On the other hand, kinetic measurements show that the effective OH uptake coefficient, $\gamma_{eff}$, for 3-MGA-AS particles (0.99 ± 0.05) is smaller than that for 3-MGA particles (2.41 ± 0.13) by about a factor of ~2.4. A smaller reactivity observed in

3-MGA-AS particles might be attributed to a higher surface concentration of water molecules, and the presence of ammonium and sulfate ions, which are chemically inert to OH radicals, at the particle surface. This could lower the collision probability between the 3-MGA and OH radicals, resulting in a smaller overall reaction rate. Our results suggest that inorganic salts likely alter the overall heterogeneous reactivity of organic compounds with gas-phase OH radicals rather than reaction

mechanisms in well-mixed aqueous organic–inorganic droplets at a high humidity (i.e. 85% RH). It also acknowledges that the effects of inorganic salts on the heterogeneous reactivity could vary greatly, depending on the particle composition and environmental conditions (e.g. RH and temperature). For instance, at lower relative humidities, aqueous 3-MGA–AS droplets likely become more concentrated and more viscous before efflorescence, possibly giving rise to diffusion limitation

during oxidation under relatively dry or cold conditions. Further studies on the effects of inorganic salts on the diffusivity of the species under different relative humidities within the organic–inorganic particles are also desirable to better understand the role of inorganic salts in the heterogeneous reactivity of organic compounds.

Our results suggest that inorganic salts likely alter the overall heterogeneous reactivity of organic compounds with gas-phase OH radicals rather than reaction mechanisms in well-mixed aqueous

~~organic-inorganic particles.~~

## 1. Introduction

Atmospheric particles are chemically complex and are comprised of a large variety of organic compounds, inorganic salts and water. Organic compounds contribute a significant mass fraction (20-90 %) of atmospheric particles (Kanakidou et al., 2005; Zhang et al., 2007; Jimenez et al., 2009). Laboratory and modeling studies have revealed that organic compounds present at or near the particle surface can be efficiently oxidized by gas-phase oxidants such as hydroxyl (OH) radicals, ozone ($O_3$) and nitrate radicals (Rudich et al., 2007; George and Abbatt, 2010; Kroll et al., 2015; Estillore et al., 2016; Chapleski et al., 2016). The effective OH uptake coefficient, $\gamma_{eff}$, defined as the fraction of OH collisions with organic molecules that yields a reaction of a target organic molecule, has been commonly used to describe the kinetics and reported for a variety of pure organic compounds. The $\gamma_{eff}$ in general has a value of $\geq 0.1$ and even $\geq 1$, indicating the occurrence of secondary chemistry (e.g. chain reactions induced by the hydrogen abstraction of organic molecules by alkoxy radicals) (Richards-Henderson et al., 2015). These heterogeneous oxidative processes can continuously alter the surface and bulk composition of the particles (Slade and Knopf, 2013; Li et al., 2018), and thus modify particle properties such as light extinction, hygroscopicity and cloud condensation nuclei activity (Petters et al., 2006; George et al., 2007; Lambe et al., 2007, 2009; Cappa et al., 2011; Slade et al., 2015; Slade et al., 2017). ~~These heterogenous oxidative processes can continuously alter the surface and bulk composition of the particles (Slade and Knopf, 2013; Li et al., 2018), and thus modify particle properties such as light extinction, hygroscopicity and cloud condensation nuclei activity (George et al., 2007; Lambe et al., 2007, 2009; Cappa et al., 2011).~~

While the transformation of pure organic particles has become more reasonably understood, the chemical transformation of organic particles in the presence of dissolved inorganic salts is largely unclear. Only a few laboratory studies have investigated the heterogeneous oxidation of

organic-inorganic particles (McNeill et al., 2007, 2008; Dennis-Smither et al., 2012). In those studies, hydrophobic organic compounds (e.g. oleic acid and palmitic acid) have been chosen as model compounds. Due to the hydrophobic nature of these compounds, the particles tend to be phase-separated with a thin organic layer on the surface of aqueous inorganic core (e.g. aqueous sodium chloride or ammonium sulfate (AS) phases). For these phase-separated particles, the molecular structure and orientation of organic molecules at the particle surface are observed to alter the reactive uptake of gas-phase oxidants such as $O_3$ and OH radicals. The reaction products formed from the ozonolysis of oleic acid-AS particles are very similar to those found in pure oleic acid particles (McNeill et al., 2007). These observations are consistent with the hypothesis that a thin organic layer effectively shields the aqueous inorganic core from being oxidized at the particle surface.

To date, there are still uncertainties on how salts affect the heterogeneous reactivity of organic compounds, in particular, the more oxygenated ones which exhibit moderate to high solubility in water. In this work, experiments were conducted to investigate the evolution of molecular composition of 3-methylglutaric acid (3-MGA) particles and particles containing 3-MGA and AS in an organic-to-inorganic mass ratio (water-free) (OIR) of 2 upon heterogeneous OH oxidation at a relative humidity (RH) of 85.0 % using an aerosol flow tube reactor coupled to the Direct Analysis in Real Time (DART) mass spectrometry. 3-MGA is chosen as a model compound for methyl-substituted dicarboxylic acids (**Table 1**), while AS is chosen as the model inorganic salt. The OIR of 2 used in this work is in the range of OIR commonly observed in atmospheric particles (Jimenez et al., 2009). The model system allows us to gain more insights into the physics and chemistry of heterogeneous reactions. The physical state of the particle is known to be a key factor in controlling the heterogeneous reactivity (Renbaum and Smith, 2009; Shiraiwa et al., 2011; Chan et al., 2014; Slade and Knopf, 2014; Fan et al., 2015; Marshall et al., 2018). Recent laboratory and modeling studies have shown that in addition to deliquescence and efflorescence, particles containing organic compounds and inorganic salts can undergo phase separation, depending on the particle composition

and environmental conditions such as RH and temperature (Ciobanu et al., 2009; Reid et al., 2011; Song et al., 2012ab; Zuend and Seinfeld, 2012; Qiu and Molinero, 2015; Stewart, et al., 2015; You and Bertram, 2015; Freedman, 2017; Losey et al., 2018). Since whether particles are well-mixed or phase-separated governs the surface composition of particles and thus the heterogeneous reactivity, the phase separation behavior of 3-MGA-AS particles was investigated using an aerosol flow cell coupled to an optical microscope. By assessing the molecular transformation of 3-MGA and 3-MGA-AS particles upon oxidation together with phase separation data obtained from optical microscopy measurements, the effects of AS on the heterogeneous OH kinetics and chemistry of 3-MGA are examined. More recently, we have measured the heterogeneous OH reactivity of pure 2-MGA particles, a structural isomer of 3-MGA (Chim et al., 2017a). Given their similar structures, the results of this work together with 2-MGA data might provide some new aspects on how dissolved inorganic salts would alter the heterogeneous kinetics and chemistry of methyl substituted dicarboxylic acids.

## 2. Experimental Methods

### 2.1 Heterogeneous OH oxidation of 3-MGA and 3-MGA-AS particles

The OH-initiated heterogeneous oxidation of 3-MGA and 3-MGA-AS particles were performed in an aerosol flow tube reactor at 85.0 % RH. Details of the experimental methods have been described elsewhere (Chim et al., 2017ab). In brief, the particle stream did not pass through a diffusion dryer and was directly mixed with $O_3$, oxygen ($O_2$), dry nitrogen ($N_2$), humidified $N_2$ and hexane before being introduced into the reactor. The RH within the reactor was controlled by varying the dry $N_2$ to humidified $N_2$ ratio and was measured at the inlet of the reactor. A water jacket around the reactor was used to maintain a constant temperature of 20 °C inside the reactor. In brief, an aqueous solution of 3-MGA or 3-MGA-AS in an OIR of 2 was atomized using a TSI atomizer to generate aqueous droplets. The particle stream was then mixed with $O_3$, oxygen ($O_2$), dry nitrogen ($N_2$), humidified $N_2$ and hexane, and was introduced into the reactor. The RH within the reactor was controlled by varying the dry $N_2$ to humidified $N_2$ ratio. A water jacket was used to maintain a constant

~~temperature of 20 °C inside the reactor.~~ Inside the reactor, gas-phase OH radicals were generated via the photolysis of $O_3$ using UV lamps at 254 nm. The processes can be described by the following reactions:

$$O_3 \rightarrow O(^1D) + O_2 \tag{R1}$$

$$O(^1D) + H_2O \rightarrow 2 \ OH \tag{R2}$$

The gas-phase concentration of OH radicals was controlled by varying the $O_3$ concentration introduced to the reactor and was determined by measuring the decay of hexane, a gas-phase tracer of OH radicals, using a gas chromatograph coupled with a flame ionization detector (GC-FID). The OH exposure, a quantity defined as the product of OH concentration and particle residence time (~1.3 min), can be determined by the following equation (Smith et al., 2009): ~~The OH exposure, a quantity defined as the product of OH concentration and particle residence time, can be determined by the following equation (Smith et al., 2009):~~

$$OH \ exposure = -\frac{\ln\left(\frac{[Hex]}{[Hex]_0}\right)}{k_{Hex}} = \int_o^t [OH] dt \tag{Eqn. 1}$$

where $k_{Hex}$ is the rate constant for the reaction of OH radicals with hexane ($5.2 \times 10^{-12}$ cm³ molecule⁻¹ s⁻¹), $[Hex]_0$ is the initial hexane concentration entering the reactor, and $[Hex]$ is the concentration of hexane leaving the reactor. The OH exposure was varied from 0 to a maximum of $4.06 \times 10^{11}$ molecules cm⁻³ s in 3-MGA experiments and was varied from 0 to a maximum of $3.84 \times 10^{11}$ molecules cm⁻³ s in 3-MGA-AS experiments. The oxidation levels are equivalent to about 3 days in the atmosphere under a moderate to high level of OH concentration ($1.5 \times 10^6$ molecules cm⁻³). Since the OH exposure (and OH concentration) was determined by the in situ measurement of the decay of hexane, the impacts of RH and water uptake by particles inside the reactor on the generation and concentration of gas-phase OH radicals have been taken into account. The competitions between the heterogeneous oxidation and the gas-phase oxidation have also been considered when quantifying OH concentration. Upon exiting the reactor, the particle stream then passed through an annular Carulite catalyst denuder for $O_3$ removal and an activated charcoal

denuder for the removal of organic gas-phase species remaining in the stream. Hence, only particle-phase reaction products were analyzed. It acknowledges that Carulite catalyst denuder can slightly decrease the RH of the particle stream. However, this would not have significant effect on the reaction products analyzed, because the decrease in RH after oxidation would not significantly affect the formation of reaction products, which primarily occurred inside the reactor. Size distribution of the particles was determined by sampling a small portion of the particle stream using a scanning mobility particle sizer (SMPS, TSI) after oxidation. Size distribution of the particles was determined by sampling a small portion of the particle stream using a scanning mobility particle sizer (SMPS, TSI). The remaining flow was directed to a heater at 250–300 ºC to fully vaporize the particles. 3-MGA and 3-MGA-AS particles were confirmed to be fully vaporized upon heating at 250 ˚C or above by measuring the size distribution of the particles leaving the heater with the SMPS in separate experiments. The resulting gas-phase species were then directed to an ionization region, a narrow open space between the DART ionization source (IonSense: DART SVP), and the inlet of the high-resolution mass spectrometer (ThermoFisher, Q Exactive Orbitrap) (Chan et al., 2013; Nah et al., 2013).

Details of the DART operation have been described in the work of Cody et al. (2005). The DART ionization source was operated in negative ionization mode with helium (He) as the ionizing gas. The formation of gas-phase ions in the ionization region can be described by below (Cody, 2009):

$$e^- + O_2(g) \rightarrow O_2^-(g) \qquad (R3)$$

$$O_2^-(g) + M(g) \rightarrow [M - H]^-(g) + HO_2(g) \quad (R4)$$

Atmospheric $O_2$ molecules abstract the electrons ($e^-$) produced by the Penning ionization of metastable He in the DART ionization source to form anionic oxygen ions ($O_2^-$) which then react with the gas-phase species (M) to form deprotonated molecular ions ($[M-H]^-$) by proton abstraction. Previous studies have shown that the acidic proton of the carboxyl group can be abstracted by the $O_2^-$ ions to generate the $[M-H]^-$, which was observed dominantly in the mass spectra (Cheng et al.,

2015; Chim et al., 2017ab). In this work, it is likely that proton abstraction from the carboxyl group of 3-MGA and its reaction products occurred to produce the [M−H]⁻. These ions were sampled by the high-resolution mass spectrometer. Mass spectra, over a scan range from *m/z* 70–400 at a mass resolution of about 140000, were collected. The mass spectra were analyzed using the Xcalibur software (Xcalibur Software, Inc., Herndon, VA, USA). It acknowledges that thermal composition of reaction products could possibly occur during the thermal desorption process (Stark et al. 2017). The mass spectra of some organic acids and alcohols (e.g. succinic acid, ketosuccinic acid and tartaric acid) are available in the work of Chan et al. (2014), showing insignificant thermal decomposition during the DART analysis. In this study, the thermal decomposition of 3-MGA was found to be insignificant as the deprotonated molecular ion of 3-MGA is the dominant peak before oxidation in the mass spectra (**Figure 1**). We acknowledge that reactions between peroxy radicals may yield organic peroxides and oligomers, which may decompose thermally. We cannot completely rule out the possibility of such reactions, but there was no indication of any fragment ions expected from the thermal decomposition in the mass spectra. Together, these results suggest that the impact of thermal decomposition on the observed product distribution is likely insignificant. Two control experiments were conducted: one in the presence of $O_3$ without the UV light and another one in the absence of $O_3$ with the UV light on. No compositional changes were observed for 3-MGA and 3-MGA-AS particles in both control experiments, indicating that the reaction of 3-MGA with $O_3$ is insignificant and that 3-MGA is not likely to be photolyzed.

The hygroscopicity data of 3-MGA has been reported in the work of Marsh et al. (2017) with a growth factor of ~1.2 at 85 % RH. As shown by the hygroscopicity curve measured, 3-MGA particles absorb and desorb water reversibly as the RH increases or decreases, indicating that they are likely aqueous droplets prior to oxidation. Optical microscopy measurements have been carried out (**Figure S1**, *supplementary material*) and show that 3-MGA-AS particles are in a single liquid phase prior to oxidation at 85.0 % RH as the particles become phase-separated when the RH is below the separation

RH (SRH = 72.7–73.6 %) (**Figure S2**, *supplementary material*). Details of the optical microscopy measurements have been given in the *Supplementary Material.*

~~Optical microscopy measurements have been carried out (**Figure S1,** *supplementary material*) and show that 3-MGA-AS particles are in a single liquid phase prior to oxidation at 85.0 % RH as the particles become phase-separated when the RH is below the separation RH (SRH = 72.7–73.6 %) (**Figure S2,** *supplementary material*). Details of the optical microscopy measurements have been given in the *Supplementary Material.*~~ Since the particles were always exposed to high humidity and the experiments were carried out at 85.0 % RH, which is higher than the SRH, 3-MGA-AS particles are likely to be single-phase liquid droplets prior to oxidation.

## 3 Results and Discussion

### 3.1 Particle mass spectra of 3-MGA and 3-MGA-AS before and after OH oxidation

**Figure 1** shows the mass spectra of 3-MGA and 3-MGA-AS before and after OH oxidation at 85.0 % RH, respectively. For 3-MGA, a dominant peak is observed before oxidation at $m/z$ = 145 which corresponds to the deprotonated molecular ion of 3-MGA ($C_6H_9O_4^-$). After oxidation, two major product peaks evolve, corresponding to two $C_6$ functionalization products ($C_6$ hydroxyl products ($C_6H_{10}O_5$) and $C_6$ ketone products ($C_6H_8O_5$)). A few minor product peaks, such as $C_4H_5O_3^-$, $C_5H_5O_3^-$, $C_5H_7O_3^-$, $C_5H_7O_4^-$ and $C_5H_7O_5^-$, are also observed. Each of these peaks contributes less than 2.5 % of the total ion signal at the maximum OH exposure. The mass spectra of 3-MGA-AS particles in OIR = 2 are very similar to those of 3-MGA particles, except for the two inorganic sulfate peaks that originate from dissolved AS. Before oxidation (**Figure 1**), three peaks at $m/z$ = 97, $m/z$ = 145 and $m/z$ = 195 are observed, corresponding to the bisulfate ion ($HSO_4^-$), the deprotonated molecular ion of 3-MGA and $H_3S_2O_8^-$, respectively. One possibility is that $HSO_4^-$ is likely the dissolved ion from aqueous AS that became acidified by the evaporative loss of ammonia ($NH_3$) into gas phase and can be detected via direct ionization in the negative ion mode (Hajslova et al., 2011). $HSO_4^-$ has been detected when aqueous AS particles are ionized by the DART ionization source as well as when

being a reaction product formed in the heterogeneous OH oxidation of sodium methyl sulfate, sodium ethyl sulfate and methanesulfonic acid particles (Kwong et al., 2018ab). However, we do not have a clear explanation for the formation of $H_3S_2O_8^-$ which is likely an adduct of $HSO_4^-$ and $H_2SO_4$. After oxidation, the deprotonated ions of $C_6$ hydroxyl ($C_6H_9O_5^-$) and $C_6$ ketone products ($C_6H_7O_5^-$) are observed in addition to the unreacted 3-MGA. Some small product peaks are detected ($C_4H_5O_3^-$, $C_5H_5O_3^-$, $C_5H_7O_3^-$, $C_5H_7O_4^-$ and $C_5H_7O_5^-$); with each contributing less than 2.5 % of the total ion signal.

As shown in **Figure 2**, the chemical evolution in the composition of 3-MGA and 3-MGA-AS particles upon oxidation are very similar. At the maximum OH exposure, the $C_6$ hydroxyl products are the most abundant species, which accounts for 38.0−48.2 % of the total organic ion signal, followed by unreacted 3-MGA (37.3−47.9 %), and the $C_6$ ketone products (7.3−7.6 %). For 3-MGA-AS, the intensities of $HSO_4^-$ and $H_3S_2O_8^-$ remain about the same before and after OH oxidation (**Figure S3,** *supplementary material*). This could be attributed to the fact that dissolved AS does not react effectively with gas-phase OH radicals at the particle surface (George and Abbatt, 2010). In general, the same reaction products are observed for both 3-MGA and 3-MGA-AS particles after oxidation, suggesting that AS does not significantly affect the reaction pathways.

### 3.3 Oxidative Kinetics of 3-MGA and 3-MGA-AS

The normalized parent decay in 3-MGA and 3-MGA-AS particles as a function of OH exposure at 85.0 % RH is shown in **Figure 3**. For both systems, the OH-initiated decay of 3-MGA follows an exponential trend and can be fit with an exponential function to obtain an effective second order OH reaction rate constant ($k$):

$$\ln \frac{I}{I_0} = -k\,[OH]\,t \qquad\qquad \text{(Eqn. 2)}$$

where $I_0$ is the ion signals of 3-MGA before oxidation, $I$ is the ion signals of 3-MGA at a given OH exposure, [OH] is the gas-phase concentration of OH radical and $t$ is the reaction time. The fitted value of $k$ for the 3-MGA and 3-MGA-AS are $(3.26 \pm 0.065) \times 10^{-12}$ cm$^3$ molecule$^{-1}$ s$^{-1}$ and $(2.72 \pm 0.064) \times 10^{-12}$ cm$^3$ molecule$^{-1}$ s$^{-1}$, respectively. Using the fitted $k$ value, the effective OH uptake coefficient, $\gamma_{eff}$, defined as the fraction of OH collisions with particles that yields a reaction, can be computed by the following equation (Davies and Wilson, 2015):

$$\gamma_{\text{eff}} = \frac{2}{3} \frac{\rho_0 D_0 \ mfs \ N_A \ k}{M_w \ \overline{c_{OH}}} \qquad \text{(Eqn. 3)}$$

where $\rho_0$ is the density of particle, $D_0$ is the particle diameter, $mfs$ is the mass fraction of 3-MGA in the particle, $M_w$ is the molecular weight of 3-MGA, $N_A$ is Avogadro's number, and $\overline{c_{OH}}$ is the mean speed of gas-phase OH radicals. The mean surface weighted diameters prior to OH oxidation (203.0 nm for 3-MGA and 200.8 nm for 3-MGA-AS, respectively) are used in the calculation of $\gamma_{eff}$. Upon oxidation, the mean surface weighted diameter decreases from 203.0 nm to 170.7 nm for 3-MGA particles and decreases from 200.8 nm to 187.8 nm for 3-MGA-AS particles (**Figure S5**, *supplementary material*). The decrease in the particle diameter upon oxidation is likely attributed to the formation and volatilization of fragmentation products and the associated evaporative loss of water molecules. Vaden et al. (2011) have discussed that evaporation of highly viscous particles is likely independent of particle size distribution and is unlikely to be significantly influence the overall evaporation behavior. As the study of Vaden et al. (2011) focused on highly viscous particles while the focus of this study is more liquid-like particles, their results may not be applicable in our study. Since 3-MGA-AS particles are more liquid-like particles, the evaporate rate would scale with the total surface area of the polydisperse particle population. Since the spread of the polydisperse particle population is small in this work, the size change is not likely substantial with regard to determining $\gamma_{eff}$ as the total particle surface area did not change dramatically. In the work of Meng and Seinfeld (1996), the mixing timescales of volatile/semi-volatile species are evaluated. Although it was suggested by the study that the timescales may increase with increasing particle size, the difference may not be that significant in our study, as the span of the polydisperse particles is much smaller than the difference between coarse particles and fine particles used in Meng and Seinfeld (1996). We thus

postulate that the spread of particle size and the mixing timescale would not play a substantial role in the evaporation of fragmentation products during oxidation. As the change in particle size upon oxidation is not very significant, the change in particle diameter was not accounted for in the $\gamma_{eff}$ calculation. The $\gamma_{eff}$ may thus be considered as an initial uptake coefficient (Chim et al., 2018). We acknowledge that the spread of particle size could potentially affect the uncertainty and determination of $\gamma_{eff}$, but we could not quantify it since the particles are polydisperse in our study. Future investigations can be carried out to measure the $\gamma_{eff}$ for both monodisperse (size-selected) and polydisperse particle populations. The $\gamma_{eff}$ assembled from different monodisperse particle sizes can be compared with that obtained from polydisperse populations using the surface-weighted mean diameter in order to assess how the spread and uncertainty in the particle size distribution of polydisperse particle populations affect the determination of $\gamma_{eff}$. ~~Before oxidation, the diameter of 3-MGA and 3-MGA-AS particles were measured to be 203.0 nm and 200.8 nm, respectively.~~ The *mfs* values were obtained from equilibrium composition calculations using the Aerosol Inorganic-Organic Mixtures Functional groups Activity Coefficients (AIOMFAC) model available online (https://aiomfac.lab.mcgill.ca) (**Table 1**) (Zuend et al., 2008; Zuend et al., 2011). Based on the composition (i.e. *mfs*), the densities of 3-MGA and 3-MGA-AS particles were estimated using the volume additivity rule with the density of water, 3-MGA and AS (Chim et al., 2017a). Using Eqn. 3, the $\gamma_{eff}$ for 3-MGA and 3-MGA-AS are calculated to be 2.41 ± 0.13 and 0.99 ± 0.05, respectively. The value of $\gamma_{eff}$ for 3-MGA particles is larger than that of 3-MGA-AS particles by about 2.4 times. ~~The values of $\gamma_{eff}$ for 3-MGA-AS particles is smaller than that of 3-MGA particles by about 59 %.~~ One possible explanation is that the mass fraction of 3-MGA in 3-MGA-AS particles (*mfs* = 0.344) is smaller than that in 3-MGA particles (*mfs* = 0.707) at 85.0 % RH before oxidation (**Table 1**), likely resulted from the presence of AS and the concomitant increase in particle hygroscopicity. A simple analysis shows that the surface coverage of 3-MGA in 3-MGA particles and 3-MGA-AS particles are roughly estimated to be 51.4% and 21.6%, respectively (*Supplementary material*). A smaller surface concentration of 3-MGA in 3-MGA-AS particles might reduce the collision

probability between 3-MGA and gas-phase OH radicals at the air-particle interface and thus lower the overall reactivity, as compared to 3-MGA particles. It should acknowledge that dissolved inorganic ions (e.g. $SO_4^{2-}$) may not be homogeneously distributed in the droplets with reference to the work of Jungwirth et al. (2003) and Jungwirth and Tobias (2006). Furthermore, the surface activity of 3-MGA is not known and slight surfactant behavior could drastically alter the surface concentration. Thus the numbers presented here are to serve as a first approximation illustrating the possible effect of AS addition on the surface coverage of 3-MGA. Further investigations on the surfactant properties of 3-MGA and molecular dynamic simulation would be useful to better understand the surface composition of both 3-MGA and 3-MGA–AS particles.

A similar result has also been observed for the OH oxidation with methanesulfonic acid (MSA) reported in the literature. Mungall et al. (2017) have investigated the heterogeneous OH oxidation of MSA-AS particles with a mass fraction of MSA = 0.16 at 75 % RH. The $\gamma_{eff}$ was reported to be 0.05 ± 0.03, which is smaller than that of pure MSA particles ($\gamma_{eff}$ = 0.45 ± 0.14) measured at a slightly higher RH (90 %) (Kwong et al., 2018b). The results obtained in this work and in the literature suggest that for a given RH, inorganic salts (e.g. AS) might lower the heterogeneous reactivity of organic compounds toward gas-phase OH radicals due to the smaller surface concentration of 3-MGA resulting from the presence of AS and concomitant increase in water uptake. It acknowledges that ammonium ($NH_4^+$) and sulfate ($SO_4^{2-}$) ions, which are chemically inert to OH radicals, present at or near the surface could lower the overall reaction rates by reducing the surface concentration of organic compounds. However, the additional effects of $NH_4^+$ and $SO_4^{2-}$ ions on the surface activity and configuration of organic molecules, which could play a role in determining the heterogeneous activity, are not yet well understood and warrant further investigations.

Kinetic measurements show that $\gamma_{eff}$ for both 3-MGA and 3-MGA-AS particles are close to or greater than one. This indicates that more than one 3-MGA molecule is reacted away per OH radical

collision with the particle surface, suggesting the occurrence of secondary chemistry in the particle phase. In the following sections, reaction mechanisms are tentatively proposed and discussed to explain the formation of major products detected in the particle mass spectra and the reaction pathways likely responsible for the secondary chemistry.

### 3.4 Reaction Mechanisms

As shown in **Figure 2**, the reaction products observed in 3-MGA and 3-MGA-AS particles are about the same upon oxidation. A generalized reaction scheme is thus proposed for both systems based on well-known particle-phase reactions previously published in the literature (Russell, 1957; Bennett and Summers, 1974; George and Abbatt, 2010). As shown in **Scheme 1,** the OH oxidation with 3-MGA can be initiated by the hydrogen abstraction at three different carbon sites: tertiary backbone carbon site (**Path A),** secondary backbone carbon site (**Path B**) and the primary carbon site of the branched methyl group (**Path C**). Depending on the initial OH reaction site, a variety of reaction products can be formed and broadly classified into two groups: functionalization and fragmentation products.

### 3.4.1 Functionalization products

At the first oxidation step, a hydrogen atom is abstracted from a 3-MGA molecule by an OH radical, forming an alkyl radical that reacts quickly with an $O_2$ molecule to form a peroxy radical. The major $C_6$ hydroxyl ($C_6H_{10}O_5$) and $C_6$ ketone ($C_6H_8O_5$) products can be generated from the self-reactions of two peroxy radicals through the Russell mechanism (**R1**) and/or Bennett-Summers reactions (**R2**). Alternatively, the self-reactions of two peroxy radicals can form two alkoxy radicals which can then abstract hydrogen atoms from the neighboring organic molecules (**R3**) to form $C_6$ hydroxyl products or react with $O_2$ molecules (**R4**) to form $C_6$ ketone products. However, when the hydrogen abstraction occurs at the tertiary carbon site (**Scheme 1, Path A**), only the $C_6$ hydroxyl product can be formed because only a hydroxyl group can be added to the tertiary carbon site. Depending on

initial reaction site, structural isomers of these $C_6$ hydroxyl and ketone products are likely formed during oxidation.

### 3.4.2 Fragmentation products

The fragmentation products can be generated from the decomposition of alkoxy radicals (**R5**). For instance, when the initial hydrogen abstraction occurs at the secondary carbon site (**Scheme 1, Path B**), the decomposition of the secondary alkoxy radical can yield either a $C_2$ ($C_2H_2O_3$) or a $C_5$ fragmentation product ($C_5H_8O_3$). On the other hand, a $C_4$ fragmentation product ($C_4H_6O_3$) can be yielded from the decomposition of the alkoxy radical formed at the tertiary carbon site (**Scheme 1, Path A**) while oxidation at the primary carbon site (**Scheme 1, Path C**) can yield a $C_3$ fragmentation product ($C_3H_4O_3$). For both 3-MGA and 3-MGA-AS, the ion signal intensity of fragmentation products is small (**Figure 1**). For example, only a small signal of $C_4$ fragmentation product ($C_4H_6O_3$), which is formed from the oxidation at the tertiary carbon site (**Scheme 1, Path A**), is detected. It contributes less than 2% of the total ion signal at the maximum OH exposure. The observed low abundances of fragmentation products could be explained by their higher volatilities (**Tables S1 and S2,** *supplementary material*) and some (e.g. $C_5H_8O_5$) may be explained by the preference on initial OH reaction site as discussed below. It is noted that for 3-MGA-AS particles, the presence of AS increases the activity coefficients of fragmentation products in the particle phase based on the thermodynamic model predictions and thus increases their volatilities in general (**Table S3,** *supplementary material*).

### 3.4.3 Large $C_6$ hydroxyl-to-$C_6$ ketone product ratio: Implications for secondary chemistry

From the particle composition data, a large $C_6$ hydroxyl-to-$C_6$ ketone product ratio is observed. At the maximum OH exposure, the relative abundance of $C_6$ hydroxyl products is about 5.0–6.6 times that of $C_6$ ketone products for both 3-MGA and 3-MGA-AS particles. We acknowledge that although the ionization efficiencies are not corrected for these products in this study, the ionization efficiency of

$C_4$ hydroxyl products is found to be about the same or even lower than that of $C_4$ ketone products during the DART ionization processes (Chan et al., 2014). The abundance of $C_6$ hydroxyl products might be even larger than that of $C_6$ ketone products after correcting their ionization efficiencies, supporting the statement above. One possible explanation for the dominance of $C_6$ hydroxyl products

is that the OH abstraction may preferentially occur at the tertiary carbon site (**Scheme 1, Path A**) since the tertiary alkyl radicals are more stable than secondary and primary alkyl radicals (Cheng et al., 2015). Only an addition of a hydroxyl group at the tertiary carbon site is allowed via the alkoxy or peroxy radical reactions.

Another possibility is that the branched methyl group may sterically hinder the two peroxy radicals from arranging into a cyclic tetroxide intermediate, which is essential for the formation of hydroxyl and ketone functionalization products through the Russell and the Bennett-Summers mechanisms (Cheng et al., 2015). Alternatively, alkoxy radicals are more likely formed through the self-reaction of two peroxy radicals and can react with neighboring organic molecules (e.g. unreacted 3-MGA) by

intermolecular hydrogen abstraction to form the $C_6$ hydroxyl products. Furthermore, as proposed by Peeters et al. (2004) and Vereecken and Peeters (2009), the strong hydrogen bonding among the two terminal carboxyl groups might lower the decomposition rate of the alkoxy radical. ~~Furthermore, as proposed by Petters et al. (2004) and Vereecken and Peeters (2009), the strong hydrogen bonding among the two terminal carboxyl groups might lower the decomposition rate of the alkoxy radical.~~

This could increase the competitiveness of the intermolecular hydrogen abstraction by the alkoxy radicals. It is worthwhile to note that the intermolecular hydrogen abstraction can regenerate an alkyl radical and eventually produce peroxy radicals. These peroxy radicals can react again with other peroxy radicals to regenerate alkoxy radicals. This allows the chain reactions to propagate and increases the overall reactivity (i.e. more than one 3-MGA molecule can be reacted per initial OH

collision via secondary chemistry and, thus $\gamma_{eff}$ is larger than one). Overall, the alkoxy radical chemistry, originating from the OH abstraction at the tertiary carbon site at the first oxidation step

(**Scheme 1, Path A**) is likely the important reaction pathway for the OH reactions with 3-MGA.

## 4. Conclusions and Atmospheric Implications

Atmospheric particles can keep colliding with gas-phase oxidants, allowing continuous oxidation to occur at or near the particle surface. To better understand how dissolved inorganic salts affect the heterogeneous chemistry and kinetics of organic compounds with gas-phase OH radicals, we investigated the kinetics, products and mechanisms of particles comprised of 3-MGA and 3-MGA-AS in an OIR of 2 upon heterogeneous OH oxidation at 85.0 % RH. Optical microscopy measurements for the detection of phase separation reveal that 3-MGA-AS particles exhibit a single liquid phase prior to oxidation. Same major reaction products are formed as a result of ~~heterogenous~~heterogeneous OH oxidation with both 3-MGA and 3-MGA-AS particles. These data suggest that the presence of aqueous AS does not significantly affect the formation pathways of major reaction products. On the other hand, in the presence of AS, the heterogeneous reactivity of 3-MGA toward gas-phase OH radicals is slower in 3-MGA-AS particles compared to that in 3-MGA particles. It is likely attributed to a lower concentration of 3-MGA at the surface of 3-MGA-AS particles relative to 3-MGA particles, reducing the collision probability between 3-MGA and gas-phase OH radicals. The results from this work and the literature suggest that the presence of dissolved salts could reduce the overall heterogeneous reactivity of organic compounds with gas-phase OH radicals at the surface by lowering the surface concentration of organic compounds at a given RH and temperature. Until recently, the kinetic parameters (e.g. $\gamma_{eff}$) reported in the literature were mostly measured based on experiments with salt-free organic particles. The chemical lifetime of organic compounds or chemical tracers against heterogeneous OH reaction in the atmosphere could be longer than expected when the salts are present. Further investigations on how the amount and types of inorganic salts alter heterogeneous kinetics and chemistry are highly desirable.

Over the past decade, laboratory and modeling studies have demonstrated that atmospheric particles

can undergo phase separation and exhibit different morphologies, which play a role in many atmospheric processes. For example, the inhomogeneous distribution of inorganic and organic species within phase-separated particles can affect the reactive uptake of gas-phase species (e.g. $N_2O_5$) (Gaston et al., 2014) and water uptake of organic-inorganic particles (Chan and Chan, 2007; Zuend and Seinfeld, 2012; Hodas et al., 2015). It is still unclear how the occurrence of liquid-liquid phase separation alters the heterogeneous reactivity of organic-inorganic particles over time. As shown in **Figure S2 (***supplementary material***)**, 3-MGA-AS particles become phase separated when the RH is below the SRH (72.7–73.6 %). The phase-separated particles might exhibit different reactivity compared to those in a single liquid phase investigated in this work since the surface composition of the particles are different in these two phases. Furthermore, there is a possibility that the phase separation behavior (e.g. SRH) of the particles may change in response to the change in the particle composition over time. ~~Furthermore, the phase separation behavior (e.g. SRH) of the particles can change in response to the change in the particle composition during oxidation over time.~~ Although the phase of oxidized 3-MGA-AS particles has not been determined experimentally in this work, the overall $\langle O/C \rangle$ is found to increase slightly from 0.67 to ~0.75 (*supplementary material*), and the SRH is expected to decrease slightly after oxidation (Bertram et al., 2011; Song et al., 2012b; You et al., 2013, 2014). Since the experimental RH inside the reactor was fixed at 85.0 %, it is very likely that 3-MGA-AS particles remain in a single liquid phase state during oxidation. During oxidation, the degree of aerosol oxidation state (e.g. expressed by $\langle O/C \rangle$) typically increases due to the formation of more oxygenated reaction products and, consequently, the SRH is expected to decrease. As shown in **Figure 4**, it is hypothesized that initially phase-separated particles might transition to a homogeneous, single liquid phase state, depending on the extent of oxidation and the environmental thermodynamic conditions. Hence, it is of interest to investigate how the phase separation characteristics of organic–inorganic particles change in response to a change in the composition upon oxidation (Slade et al., 2015; Slade et al., 2017). ~~Hence, it is of interest to investigate how the phase separation characteristics of organic-inorganic particles change in response~~

~~to a change in the composition upon oxidation.~~ Moreover, with respect to future work, it would be interested in understanding the dynamic interplay between the particle composition, heterogeneous reactivity, liquid-liquid phase separation and effects on particle morphology under different environmental conditions and extents of oxidation.

**Data availability**

The underlying research data are available upon request from the corresponding author (mnchan@cuhk.edu.hk).

**Author contributions**

HKL, SMS, and MNC designed and ran the experiments. HKL, SMS, and MNC prepared the manuscript. All co-authors provided comments and suggestions to the manuscript.

**Acknowledgements**

This work is supported by the Hong Kong Research Grants Council (HKRGC) Project ID: 2191111 (Ref 24300516). We would like to thank Kevin Wilson for his insightful comments.

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

**Table 1.** Chemical structure, properties, effective heterogeneous OH rate constant, effective OH uptake coefficient of 3-methylglutaric acid (3-MGA) and 3-MGA mixed with ammonium sulfate (AS) in an organic-to-inorganic dry mass ratio (OIR) = 2.

| Chemical structure | | |
|---|---|---|
| |  | |
| **Chemical formula** | $C_6H_{10}O_4$ | |
| **O/C ratio** | 0.67 | |
| **H/C ratio** | 1.67 | |
| | **3-MGA** | **3-MGA-AS (OIR =2)** |
| **Separation RH (SRH)** | – | 72.7–73.6 % |
| **Mass fraction at 85 % RH** | | |
| **3-MGA** | 0.707 | 0.344 |
| **AS** | 0 | 0.172 |
| **$H_2O$** | 0.293 | 0.484 |
| **Effective second order heterogeneous OH rate constant, $k$ ($\times 10^{-12}$ cm$^3$ molecule$^{-1}$ s$^{-1}$)** | 3.26 ± 0.065 | 2.72 ± 0.064 |
| **Effective OH uptake coefficient, $\gamma_{eff}$** | 2.41 ± 0.13 | 0.99 ± 0.05 |

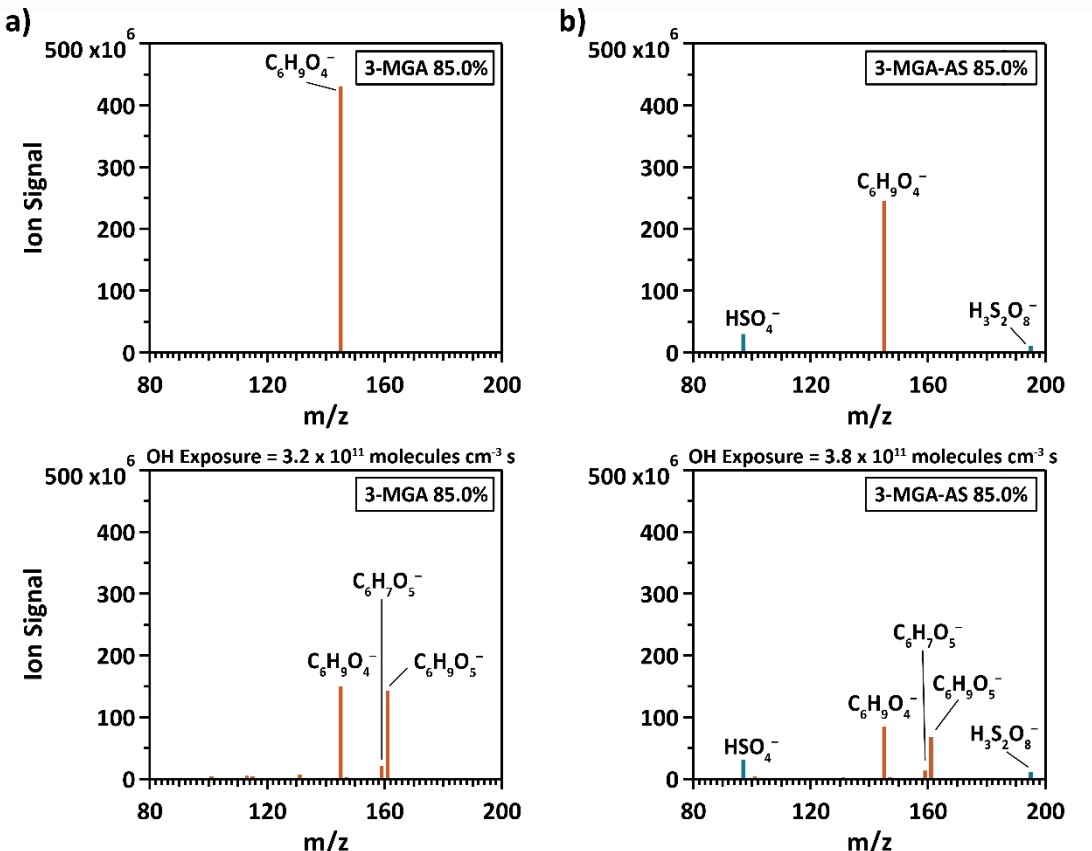

**Figure 1.** The particle mass spectrum of a) 3-MGA and b) 3-MGA-AS before (upper panels) and after (lower panels) heterogeneous OH oxidation at 85.0 % RH, respectively. Color scheme: brown = organic species, blue = inorganic species.

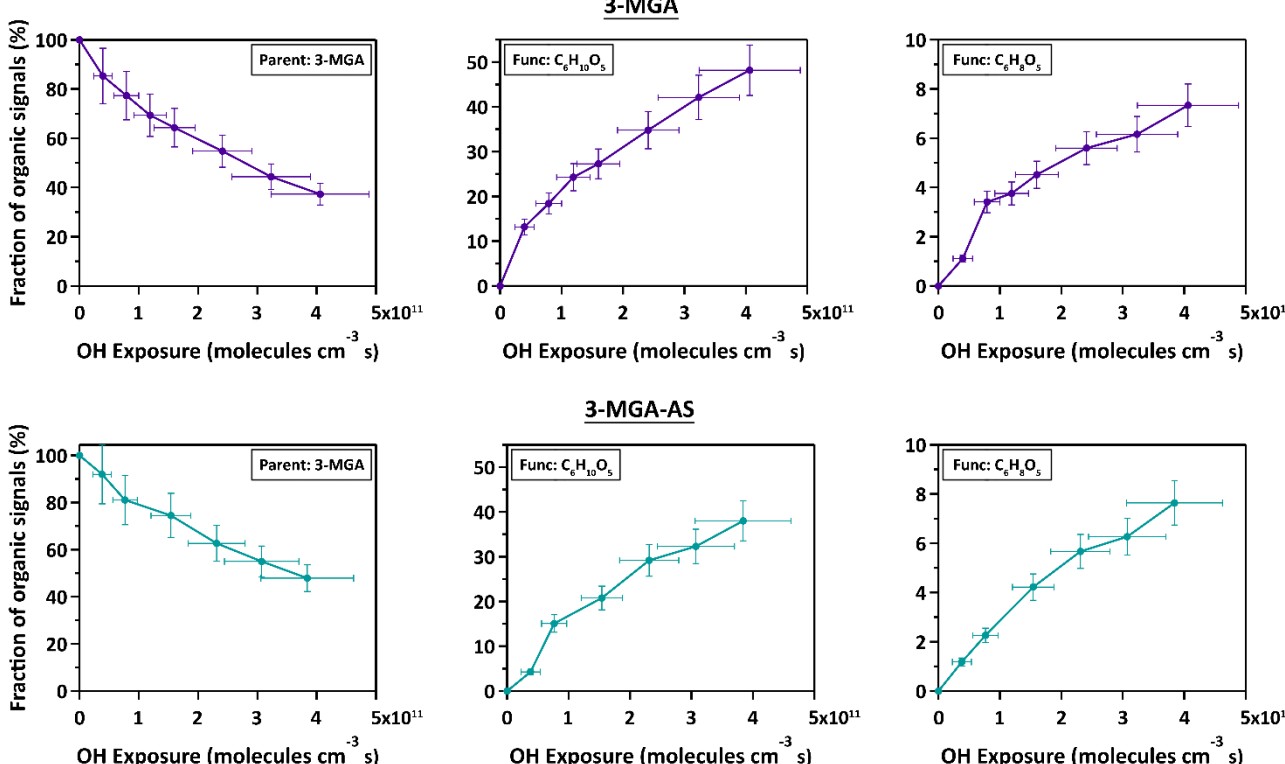

**Figure 2.** Upper panels: The fraction of the organic ion signal attributed to the parent 3-MGA, the major $C_6$ hydroxyl and $C_6$ ketone products of 3-MGA particles during the heterogeneous OH oxidation shown against OH exposure. Lower panels: analogous to the above, but for the case of mixed 3-MGA-AS particles.

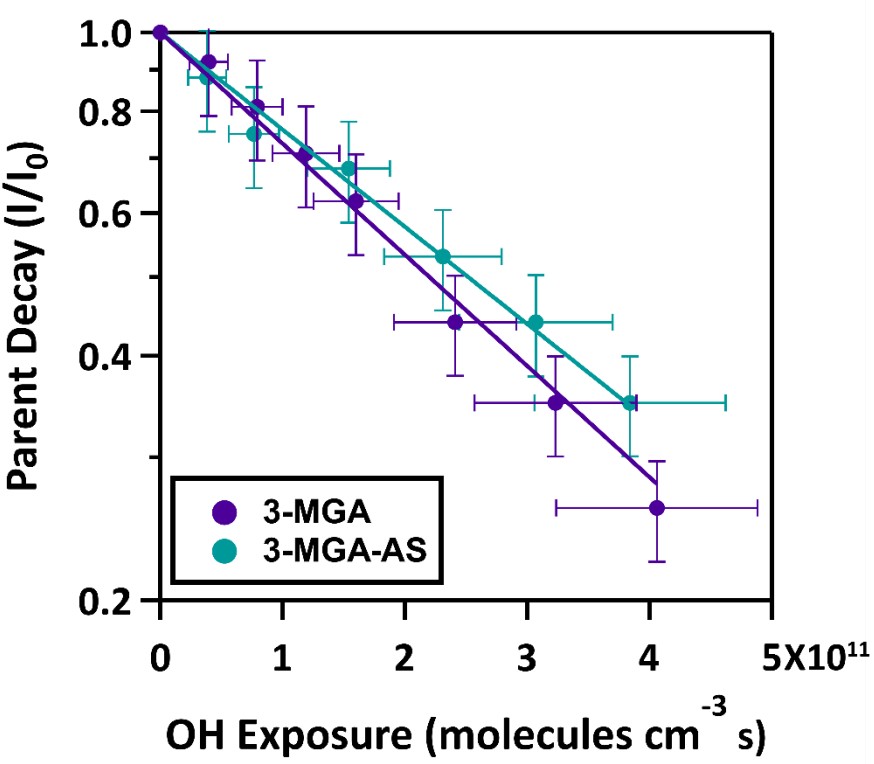

**Figure 3**. The normalized parent decay for the heterogeneous OH oxidation of 3-MGA and 3-MGA-AS particles at 85.0 % RH. Note the logarithmic scale of the ordinate.

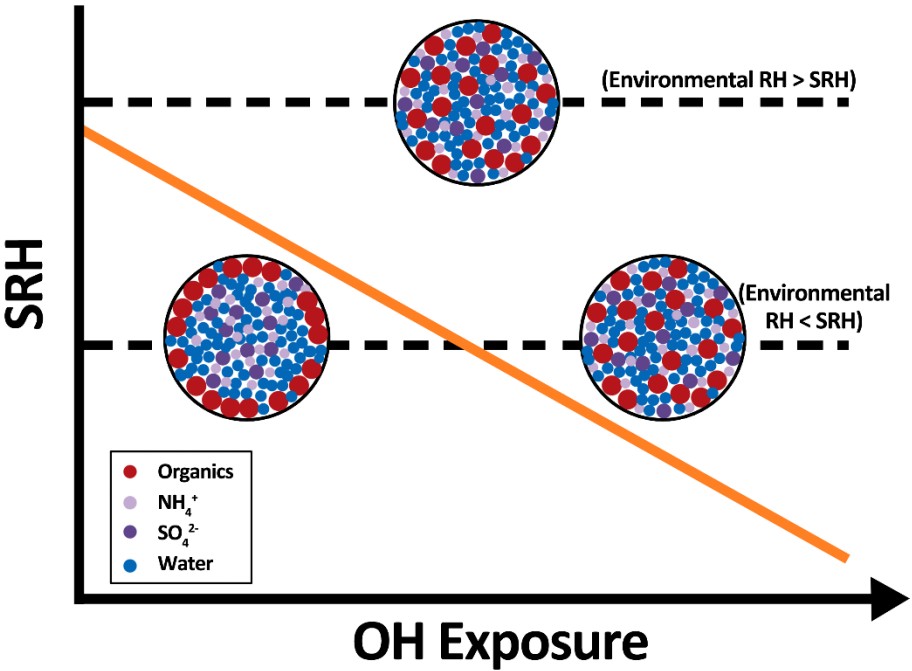

**Figure 4**. A simplified diagram illustrating the change in separation relative humidity (SRH, orange solid line) and phase composition of droplets containing inorganic salts and organic compounds (single liquid phase vs. liquid-liquid phase separated) upon heterogeneous oxidation at two different environmental RH.

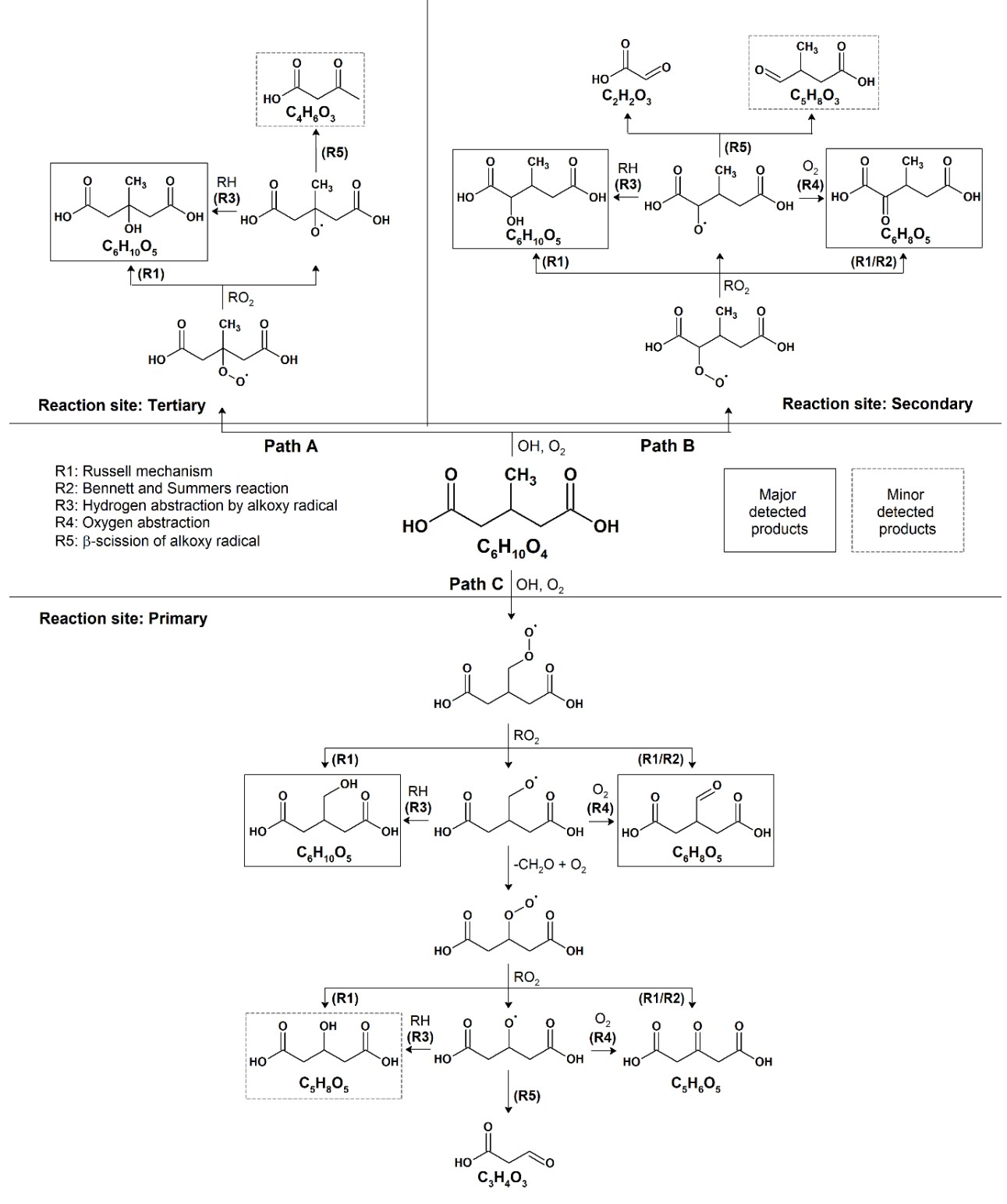

**Scheme 1.** Proposed reaction mechanisms for the heterogeneous OH oxidation of 3-MGA and 3-MGA-AS particles.

**TOC.**

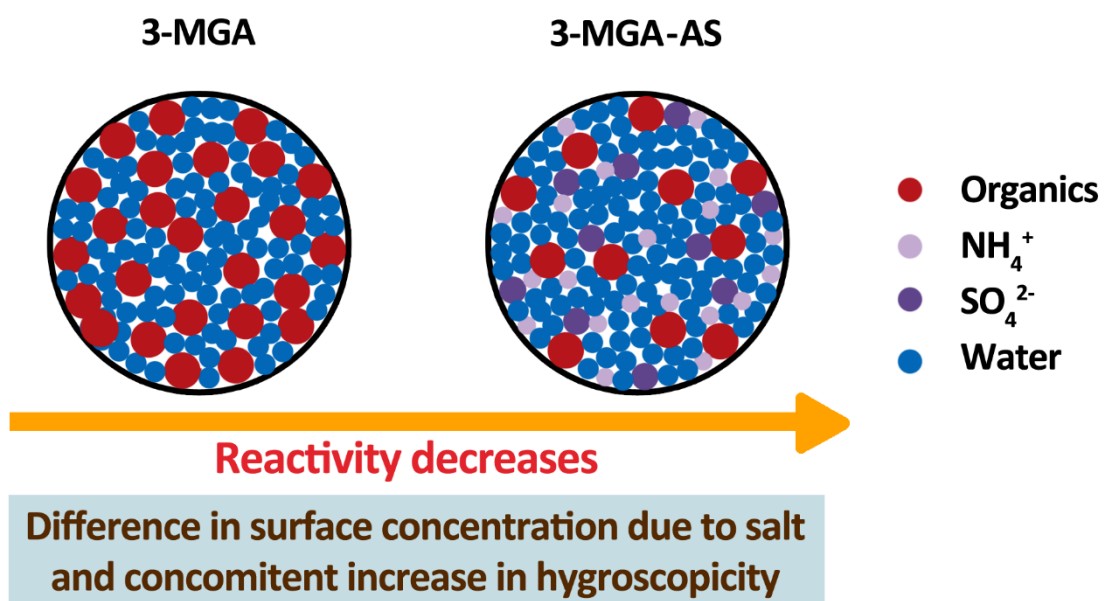