# Peer review of "Effects of Inorganic Salts on the Heterogeneous OH Oxidation of Organic"

_Atmospheric Chemistry and Physics, 2019_

## Referee Comment (RC1) · Anonymous Referee #1 · 3 Apr 2019

This study examines the impact of the presence of hygroscopic ammonium sulfate (AS) on the heterogeneous OH oxidation of 3-methylglutaric acid (3-MGA) particles at 85% relative humidity (RH). Complementary microscopy measurements show that 3-MGA-AS particles are in a single liquid phase prior to oxidation at high RH. The effective OH uptake coefficient for 3-MGA-AS particles is determined to be smaller than that for 3-MGA particles by about a factor of $\sim$2.4. The OH oxidation products are found to be the same for both particle systems investigated using Direct Analysis in Real Time (DART). The observation of smaller reactivity for 3-MGA-AS particles is explained by a higher surface concentration of water molecules and ammonium and sulfate ions, which are chemically inert to OH radicals. This may lower the collision probability between the

3-MGA and OH radicals, resulting in a smaller overall reaction rate but similar reaction products.

The topic of this study fits well within the scope of Atmospheric Chemistry and Physics. Indeed, the impact of salts on the heterogeneous oxidation kinetics of organic particulate mass involving OH radicals has not been much studied. This manuscript reads well and I have only minor revisions to suggest before publication of this study.

For making interpretation of the data easier, it would be beneficial to mention some experimental parameters regarding the flow reactor OH exposure studies. For example, after atomization, the particles are likely in a solid/crystalline phase state. How long was the particle residence time in the flow reactor? Is it assumed that the particles were completely deliquesced for the entire OH exposure time (residence time)? In other words, did the particles have sufficient time to adjust to RH? How was RH controlled and maintained in the flow reactor? RH and water uptake may impact OH concentration? How water soluble is 3-MGA? Comparison to similar soluble species and corresponding hygroscopicity factor (and growth) should be mentioned to support the case that the particles are homogeneously mixed under the conditions in the flow reactor experiment.

The size of the particles is a crucial input parameter when deriving the uptake coefficient. It is not clear if the particle size distribution was measured under dry or humidified conditions? It is also not clear if the size distribution was determined before or after OH oxidation? If acquired after oxidation, one would need to show that the particle sizes did not change upon oxidation. Lastly, which particle diameter was chosen to calculate the uptake coefficient? Does the spread and uncertainty in the size distribution contribute significantly to the uncertainty in the reactive uptake coefficient?

The reactivity between 3-MGA and 3-MGA-AS particles varies by a factor of 2.4. The authors suggest that this is due a different surface concentration of 3-MGA and corresponding difference in collision flux among these particle systems. I am wondering why

the authors do not show this using, e.g., the resistor model? In the derivation of the reactive uptake coefficient, one normalizes with the collision flux. Assuming a surface reaction, one may be able to verify, if indeed the change in collision flux can explain the difference in the determined uptake coefficients. If added to the manuscript, this would significantly elevate the results of this manuscript.

Specific comments:

p. 3, l. 5: Please add the studies by Petters et al., GRL, 2006, Slade et al., ACP, 2015 and Slade et al., GRL, 2017 who studied the OH oxidation of organic and inorganic/organic particles and its effect on hygroscopicity.

p. 4, l. 24: As mentioned in general comments, more details on SMPS measurements are required.

p. 9, Eqn. 3: Please discuss particle diameter applied.

p. 9, l. 15-20 and p. 10, l. 1-8): It may be a too simple assumption that the ions are homogeneously distributed in small droplets. Please refer to Jungwirth and Tobias, Chem. Rev., 2006; Jungwirth et al., Chem. Phys. Lett., 2003 and subsequent studies. E.g. $SO_4^{2-}$ is likely not found at the particle surface but in the interior in contrast to the schematics in Fig. 4. Also, can it be ruled out that 3-MGA may show surfactant behavior? Even slight surfactant behavior could alter the surface concentration drastically.

p. 12, l. 24: Citations Petters et al. (2004) and Vereecken and Peeters (2009) are not given in bibliography and may be wrong as well?

p. 14, l. 15: "...over time.". Please add reference. Who has shown this?

p. 14, l. 24: "...upon oxidation.". Please cite here the studies by Slade et al., ACP, 2015 and Slade et al., GRL, 2017.

References:

Petters, M., Prenni, A. J., Kreidenweis, S. M., DeMott, P. J., Matsunaga, A., Lim, Y. B., Ziemann, P. J., Chemical aging and the hydrophobic-to-hydrophilic conversion of carbonaceous aerosol, Geophys. Res. Lett., 33, L24806, 2006.

Slade, J. H., Thalman, R., Wang, J., Knopf, D. A., Chemical aging of single and multi-component biomass burning aerosol surrogate particles by OH: implications for cloud condensation nucleus activity, Atmos. Chem. Phys., 15, 10183–10201, 2015.

Slade, J. H., M. Shiraiwa, A. Arangio, H. Su, U. Pöschl, J. Wang, Knopf, D. A., Cloud droplet activation through oxidation of organic aerosol influenced by temperature and particle phase state, Geophys. Res. Lett., 44, 1583–1591, 2017.

Jungwirth, P., Tobias, D. J., Specific Ion Effects at the Air/Water Interface, Chem. Rev., 106, 4, 1259-1281, 2006.

Jungwirth, P., Curtis, J. E., Tobias, D. J., Polarizability and aqueous solvation of the sulfate dianion, Chem. Phys. Lett., 367, 5-6, 704-710, 2003.

---

## Referee Comment (RC2) · Anonymous Referee #2 · 21 May 2019

Overview

This study reports differences in the heterogeneous OH oxidation kinetics and particle-phase products of representative pure-component organic aerosol and mixed organic-inorganic aerosol under conditions of particle deliquescence. The authors report the reactive uptake coefficients between OH + methylglutaric acid and OH + methylglutaric acid/ammonium sulfate aerosol measured using an oxidation flow cell coupled with a direct analysis in real time (DART) mass spectrometer and conclude that while oxidation products are similar between the two aerosol systems, the uptake kinetics are significantly slower in the case of the methylglutaric acid/ammonium sulfate aerosol

mixture. Overall, the manuscript is well written, the topic is of interest, and the study appears to be sound/not overstated. I recommend publication in ACP after the authors respond to the following comments.

Comments

Page 2, line 4: Be careful not to understate the role of dissolved salts – this study was performed on deliquesced particles at 85% relative humidity. Because of the hygroscopicity of AS, 3-MGA-AS will deliquesce at a lower relative humidity than pure 3-MGA particles. This may not significantly affect the reaction mechanism itself, but diffusion of reactants from the particle bulk to the surface may be quite different at a different relative humidity, thus the extent of reaction I would expect depends also on the diffusivity of the reactants, which may be more important under more relevant daytime relative humidity (when [OH] peaks in the real environment). Written like this suggests the inorganic component has no influence on the reaction.

Page 5, lines 10-12: Presumably, quantification of [OH] was done prior to addition of the aerosol particles to the flow cell? The reported second-order rate coefficients for heterogeneous OH oxidation of 3-MGA and 3-MGA-AS (2.72-3.26$\times$10-12 cm3 molecule-1 s-1) are competitive with that for hexane (5.21$\times$10-12 cm3 molecule-1 s-1) (Atkinson, 2003). Please specify. If mixed, how would this affect the determination of [OH]?

Page 5, lines 21-23: Please indicate where the relative humidity measurement was made in the setup. Do you expect the particles that get sampled through the inlet of the DART instrument to be at a different relative humidity than when they were oxidized? From personal experience, Carulite catalyst can decrease the relative humidity. Please comment on potential variations in the relative humidity as part of the experimental setup and whether it has an effect on the products analyzed.

Page 6, lines 1-4: It is known that thermal desorption methods lead to inaccurate estimates of particle volatility, e.g., Stark et al. (2017) demonstrate that many organic acids and alcohols, common constituents of secondary organic aerosol, can decompose at temperatures as low as 200 °C. In that study, a significant fraction of detected compounds resulted from thermal decomposition, suggesting the detected fragments were not actually present in the atmosphere, but rather formed during decomposition in the inlet of the instrument. Please discuss potential caveats of the thermal desorption technique used here and its impact on the observed product distribution.

Page 9, lines 4-5: Please specify whether these studies were performed using mono- or polydisperse aerosol. Are the reported diameters a median value or geometric mean? Also, are the reported diameters number- or surface area-weighted? Please comment on the effect of particle size, e.g., regarding evaporation (Vaden et al., 2011) and mixing timescales of volatile/semi-volatile components (Meng and Seinfeld, 1996).

Page 9, lines 11-12: Isn't $\gamma$eff for 3-MGA-AS more than twice as small as that for 3-MGA, rather than 59%? The relative percent difference is 59%. This is a bit confusing as written. Please consider rephrasing.

Did the authors measure the aerosol size distribution after OH oxidation? If so, is there evidence of particle mass growth (functionalization) or volatilization (fragmentation)? This can be assessed simply by plotting the ratio of initial aerosol volume to final aerosol volume as a function of OH exposure. Such an analysis would be a valuable addition to the paper.

The authors provide a reasonable argument for the difference in $\gamma$eff between 3-MGA and 3-MGA-AS particles, but I would caution extension (i.e., overall applicability) of Eq. S1 to other aerosol systems at different relative humidity. Equation S1 is an over-simplification of the likely complex interactions and concentration gradients present in atmospheric aerosol. In this study, Eq. 1 simply indicates there is less 3-MGA in the 3-MGA-AS mixture to react with OH compared to the pure 3-MGA particles. However, if the particles were phase-separated or exhibited core-shell structure, e.g., at low relative humidity, does Eq. S1 indicate what is at the surface?

References

[Figure]

Atkinson, R.: Kinetics of the gas-phase reactions of OH radicals with alkanes and cycloalkanes, Atmos Chem Phys, 3, 2233-2307, DOI 10.5194/acp-3-2233-2003, 2003.

Meng, Z. Y., and Seinfeld, J. H.: Time scales to achieve atmospheric gas-aerosol equilibrium for volatile species, Atmospheric Environment, 30, 2889-2900, Doi 10.1016/1352-2310(95)00493-9, 1996.

Stark, H., Yatavelli, R. L. N., Thompson, S. L., Kang, H., Krechmer, J. E., Kimmel, J. R., Palm, B. B., Hu, W. W., Hayes, P. L., Day, D. A., Campuzano-Jost, P., Canagaratna, M. R., Jayne, J. T., Worsnop, D. R., and Jimenez, J. L.: Impact of Thermal Decomposition on Thermal Desorption Instruments: Advantage of Thermogram Analysis for Quantifying Volatility Distributions of Organic Species, Environmental Science & Technology, 51, 8491-8500, 10.1021/acs.est.7b00160, 2017.

Vaden, T. D., Imre, D., Beranek, J., Shrivastava, M., and Zelenyuk, A.: Evaporation kinetics and phase of laboratory and ambient secondary organic aerosol, Proceedings of the National Academy of Sciences of the United States of America, 108, 2190-2195, 10.1073/pnas.1013391108, 2011.

---

## Author Comment (AC1) · 29 Jun 2019

*This study examines the impact of the presence of hygroscopic ammonium sulfate (AS) on the heterogeneous OH oxidation of 3-methylglutaric acid (3-MGA) particles at 85% relative humidity (RH). Complementary microscopy measurements show that 3-MGA–AS particles are in a single liquid phase prior to oxidation at high RH. The effective OH uptake coefficient for 3-MGA–AS particles is determined to be smaller than that for 3- MGA particles by about a factor of ~2.4. The OH oxidation products are found to be the same for both particle systems investigated using Direct Analysis in Real Time (DART). The observation of smaller reactivity for 3-MGA–AS particles is explained by a higher surface concentration of water molecules and ammonium and sulfate ions, which are chemically inert to OH radicals. This may lower the collision probability between the 3-MGA and OH radicals, resulting in a smaller overall reaction rate but similar reaction products. The topic of this study fits well within the scope of Atmospheric Chemistry and Physics. Indeed, the impact of salts on the heterogeneous oxidation kinetics of organic particulate mass involving OH radicals has not been much studied. This manuscript reads well and I have only minor revisions to suggest before publication of this study.*

**We would like to sincerely thank the reviewer for his/her thoughtful comments and suggestions. Please see our responses to reviewer's comments and suggestions below.**

**Comment #1**

*For making interpretation of the data easier, it would be beneficial to mention some experimental parameters regarding the flow reactor OH exposure studies. For example, after atomization, the particles are likely in a solid/crystalline phase state. How long was the particle residence time in the flow reactor? Is it assumed that the particles were completely deliquesced for the entire OH exposure time (residence time)? In other words, did the particles have sufficient time to adjust to RH? How was RH controlled and maintained in the flow reactor? RH and water uptake may impact OH concentration? How water soluble is 3-MGA? Comparison to similar soluble species and corresponding hygroscopicity factor (and growth) should be mentioned to support the case that the particles are homogeneously mixed under the conditions in the flow reactor experiment.*

**Author Response:**

Thanks for the comments. In our experiments, aqueous droplets generated by the atomizer did not pass through a diffusion dryer and were directly mixed with gases such as humidified nitrogen, oxygen, ozone and hexane before entering the flow tube reactor. The aqueous droplets

would have sufficient time to adjust to RH (i.e. 85%) after mixing and before entering the reactor. Since the particles were always exposed to high humidity in our system, 3-MGA and 3-MGA–AS particles were always aqueous droplets for the entire OH exposure. The particle residence time in the flow tube reactor is experimentally determined to be 1.3 min.

The RH was controlled and maintained by the ratio of dry to humidified nitrogen stream being introduced into the reactor. A water jacket around the flow tube reactor was used to maintain temperature inside the reactor. In this study, gas-phase OH radicals were generated through the photolysis of ozone in the presence of water vapor. The concentration of OH radicals would be higher when RH (and the amount of water vapor) increases. Since the particles had achieved their equilibrium states before entering the reactor, we expect the water uptake of particles would not significantly alter the amount of water vapor inside the reactor and the generation of gas-phase OH radicals. Furthermore, in this study, the OH exposure was determined by the in situ measurement of the decay of hexane inside the reactor. The impacts of RH and water uptake by particles inside the reactor on the generation and concentration of gas-phase OH radicals would have been taken into account.

We have measured a solubility of 45.3 wt% in water for 3-MGA at 298K, which is highly soluble. The hygroscopicity data of 3-MGA has been shown in the work of Marsh et al. (2017) with a growth factor of ~1.2 at 85 % RH. As shown by the hygroscopicity curve measured, 3-MGA particles absorb and desorb water reversibly as the RH increases or decreases, indicating that they are likely aqueous droplets prior to oxidation. With the addition of AS, the morphology of 3-MGA–AS particles (OIR = 2) upon dehumidification also demonstrate a homogeneous single liquid phase (see the supplementary material), suggesting that 3-MGA–AS particles are likely well-mixed at high humidity (i.e. 85%). We have added the above information in the revised manuscript.

Page 5, Line 12, "In brief, the particle stream did not pass through a diffusion dryer and was directly mixed with $O_3$, oxygen ($O_2$), dry nitrogen ($N_2$), humidified $N_2$ and hexane before being introduced into the reactor. The RH within the reactor was controlled by varying the dry $N_2$ to humidified $N_2$ ratio and was measured at the inlet of the reactor. A water jacket around the reactor was used to maintain a constant temperature of 20 °C inside the reactor."

Page 5, Line 23, "The OH exposure, a quantity defined as the product of OH concentration and particle residence time (~1.3 min), can be determined by the following equation (Smith et al., 2009):"

Page 6, Line 8, "Since the OH exposure (and OH concentration) was determined by the in situ measurement of the decay of hexane, the impacts of RH and water uptake by particles inside the reactor on the generation and concentration of gas-phase OH radicals have been taken into account. The competitions between the heterogeneous oxidation and the gas-phase oxidation have also been considered when quantifying OH concentration."

Page 8, Line 7, "The hygroscopicity data of 3-MGA has been reported in the work of Marsh et al. (2017) with a growth factor of ~1.2 at 85 % RH. As shown by the hygroscopicity curve measured, 3-MGA particles absorb and desorb water reversibly as the RH increases or decreases, indicating that they are likely aqueous droplets prior to oxidation. Optical microscopy measurements have been carried out (**Figure S1**, *supplementary material*) and show that 3-MGA–AS particles are in a single-liquid phase prior to oxidation at 85.0 % RH as the particles

become phase-separated when the RH is below the separation RH (SRH = 72.7–73.6 %) (**Figure S2**, *supplementary material*). Details of the optical microscopy measurements have been given in the Supplementary Material."

**Comment #2**

*The size of the particles is a crucial input parameter when deriving the uptake coefficient. It is not clear if the particle size distribution was measured under dry or humidified conditions? It is also not clear if the size distribution was determined before or after OH oxidation? If acquired after oxidation, one would need to show that the particle sizes did not change upon oxidation. Lastly, which particle diameter was chosen to calculate the uptake coefficient? Does the spread and uncertainty in the size distribution contribute significantly to the uncertainty in the reactive uptake coefficient?*

**Author Response:**

The particle size distribution was measured under humidified conditions after leaving the reactor at each OH exposure recorded. From the particle size measurements, the surface-weighted mean diameter decreases from 203.0 nm to 170.7 nm for 3-MGA particles and decreases from 200.8 nm to 187.8 nm for 3-MGA–AS particles upon oxidation (please see the figure below). The decrease in the particle diameter upon oxidation is likely attributed to the formation and volatilization of fragmentation products and the associated evaporative loss of water molecules. We have added the change in particle diameter as a function of OH exposure (**Figure S5**, *supplementary material*) in the supporting material.

The particle size used in the calculation of the uptake coefficient (i.e. $\gamma_{eff}$) is the mean surface-weighted diameter prior to OH oxidation. As the change in particle size upon oxidation is not very significant, the initial mean surface-weighted diameter was used. We did not account for the change in particle diameter in our calculation of uptake coefficient. The uptake coefficient in this work may thus be considered as an initial uptake coefficient.

We agree with the reviewer that the spread of particle size could potentially affect the determination and uncertainty of uptake coefficient but we could not quantify it since the particles are polydisperse in this study. To address this comment, we would like to suggest that future investigations can be carried out to measure the uptake coefficients for both monodispersed (size-selected) particles and polydispered particles for the same reaction system upon oxidation. The uptake coefficients assembled from different monodisperse particle populations can be compared with that obtained from polydisperse particle populations using the mean surface-weighted diameter in order to assess how the spread and uncertainty in the size distribution of polydisperse particle affect the determination of the uptake coefficient. We have revised the manuscript to clarify these issues.

[Figure]

**Figure S5**. The change in surface-weighted mean diameter as a function of OH exposure for 3-MGA particles and 3-MGA–AS particles, respectively.

Page 10, Line 16, "The surface-weighted mean diameters prior to OH oxidation (203.0 nm for 3-MGA and 200.8 nm for 3-MGA–AS, respectively) are used in the calculation of $\gamma_{eff}$. Upon oxidation, the surface-weighted mean diameter decreases from 203.0 nm to 170.7 nm for 3-MGA particles and decreases from 200.8 nm to 187.8 nm for 3-MGA–AS particles (**Figure S5**, *supplementary material*). The decrease in the particle diameter upon oxidation is likely attributed to the formation and volatilization of fragmentation products and the associated evaporative loss of water molecules. Vaden et al. (2011) have discussed that evaporation of highly viscous particles is likely independent of particle size distribution and is unlikely to be significantly influence the overall evaporation behavior. As the study of Vaden et al. (2011) focused on highly viscous particles while the focus of this study is more liquid-like particles, their results may not be applicable in our study. Since 3-MGA-AS particles are more liquid-like particles, the evaporate rate would scale with the total surface area of the polydisperse particle population. Since the spread of the polydisperse particle population is small in this work, the size change is not likely substantial with regard to determining $\gamma_{eff}$ as the total particle surface area did not change dramatically. In the work of Meng and Seinfeld (1996), the mixing timescales of volatile/semi-volatile species are evaluated. Although it was suggested by the study that the timescales may increase with increasing particle size, the difference may not be that significant in our study, as the span of the polydisperse particles is much smaller than the difference between coarse particles and fine particles used in Meng and Seinfeld (1996). We thus postulate that the spread of particle size and the mixing timescale would not play a substantial role in the evaporation of fragmentation products during oxidation. As the change in particle size upon oxidation is not very significant, the change in particle diameter was not accounted for in the $\gamma_{eff}$ calculation. The $\gamma_{eff}$ may thus be considered as an initial uptake coefficient (Chim et al., 2018). We acknowledge that the spread of particle size could potentially affect the uncertainty and determination of $\gamma_{eff}$, but we could not quantify it since the particles are polydisperse in our study. Future investigations can be carried out to measure the $\gamma_{eff}$ for both monodisperse (size-selected) and polydisperse particle populations. The $\gamma_{eff}$ assembled from different monodisperse particle

sizes can be compared with that obtained from polydisperse populations using the surface-weighted mean diameter in order to assess how the spread and uncertainty in the particle size distribution of polydisperse particle populations affect the determination of $\gamma_{eff}$."

*Supporting material*, we have added the **Figure S5** in the *supporting material* to illustrate the change in particle diameter for 3-MGA particles and 3-MGA–AS particles upon oxidation.

**Comment #3**

*The reactivity between 3-MGA and 3-MGA–AS particles varies by a factor of 2.4. The authors suggest that this is due a different surface concentration of 3-MGA and corresponding difference in collision flux among these particle systems. I am wondering why the authors do not show this using, e.g., the resistor model? In the derivation of the reactive uptake coefficient, one normalizes with the collision flux. Assuming a surface reaction, one may be able to verify, if indeed the change in collision flux can explain the difference in the determined uptake coefficients. If added to the manuscript, this would significantly elevate the results of this manuscript.*

**Author Responses**

Thanks for the suggestion. We agree with the reviewer that a quantitative analysis could provide more insight into how change in collision flux or particle surface composition would affect the oxidative kinetics. As suggested by the reviewer, we have attempted to analyze the uptake coefficient ($\gamma_{meas} = \gamma_{eff}$) using a resistor model developed by Worsnop et al. (2002)

$$\frac{1}{\gamma_{meas}} = \frac{1}{\Gamma_{diff}} + \frac{1}{S} + \cfrac{1}{\Gamma_s + \cfrac{1}{\cfrac{S-\alpha}{S\alpha} + \cfrac{1}{\Gamma_{rxn}}}} + \frac{1}{\Gamma_{diff}^p} \qquad (\text{Eq. 1})$$

where $\Gamma_{diff}$ represents gas-phase diffusion, $\Gamma_s$ represents surface reaction, $S$ is the adsorption accommodation coefficient, $\alpha$ is the mass accommodation coefficient, $\Gamma_{rxn}$ represents chemical reaction in the particle bulk and $\Gamma_{diff}^p$ represents a diffusion-limited gradient of the reactant concentration (i.e. 3-MGA) within the particle. For heterogeneous oxidation, the gas-phase species (i.e. OH radical) are likely reacting with particle-phase species (i.e. 3-MGA) at or near the surface via a mechanism that is kinetically separable from reaction within the particle bulk or from other processes. If we assume an efficient surface reactivity, we could simplify the resistor model into (Eq. 2), which represents for the surface reaction (Worsnop et al., 2002):

$$\gamma_{\text{meas}} = \Gamma_s = \frac{4k_2^s H_s R T K_s [Y]}{\bar{c}} \qquad (\text{Eq. 2})$$

where $k_2^s$ is the second-order rate constant at the surface, $H_s$ is the Henry's law constant (M atm$^{-1}$ L$^{-1}$), $[Y]$ is the surface concentration of the species (i.e. 3-MGA), $K_s$ is the thermodynamic equilibrium constant linking the surface concentration to the bulk concentration (or activities when non-ideality is considered) and $\bar{c}$ is the average thermal speed of gas-phase OH molecules. We would like to note that this formulation (Eq. 1) has been normalized to the molecular collision rate (Worsnop et al., 2002). If we understand the reviewer's comment correctly, we

could not assess how the change in collision flux (equivalent to molecular collision) affects the determination of uptake coefficient ($\gamma_{\text{meas}} = \gamma_{\text{eff}}$). From Eq. 2, the reduced form of the resistor model for surface reaction also suggests that the uptake coefficient would depend on the surface concentration of 3-MGA (i.e. $K_s[Y]$). While some parameters required in this formulation can be obtained in this study (e.g. $\gamma_{\text{meas}}$ and $k_2^s$), some parameters (e.g. $H_s$ and $K_s$) are not well understood for 3-MGA and 3-MGA–AS particles. This might limit the use of the model for the analysis. We would also like to acknowledge that the uptake coefficient considered by Worsnop et al. (2002) as well as many other resistor models is reported from the perspective of a colliding gas-phase OH radical, whereas the one measured in this work is reported from the perspective of a 3-MGA molecule at the particle surface. Thus, the results obtained from these two approaches may not be comparable. We thus do not plan to analyze the kinetic data using the resistor model, but agree that dynamic and molecular simulations together with the experimental data would greatly help to understand how the change in collision flux and particle surface composition govern the rate of reactions.

**Specific Comments #1**

*Page 3, line 5: Please add the studies by Petters et al., GRL, 2006, Slade et al., ACP, 2015 and Slade et al., GRL, 2017 who studied the OH oxidation of organic and inorganic/organic particles and its effect on hygroscopicity.*

**Author Response:**

We have cited the studies mentioned in the revised manuscript.

Page 3, Line 12: "These heterogeneous oxidative processes can continuously alter the surface and bulk composition of the particles (Slade and Knopf, 2013; Li et al., 2018), and thus modify particle properties such as light extinction, hygroscopicity and cloud condensation nuclei activity (Petters et al., 2006; George et al., 2007; Lambe et al., 2007, 2009; Cappa et al., 2011; Slade et al., 2015; Slade et al., 2017)."

**Specific Comments #2**

*Page 4, line 24: As mentioned in general comments, more details on SMPS measurements are required.*

**Author Response:**

We have added more details on SMPS measurements and discussed the use of surface-weighted mean diameter prior to OH oxidation for the calculation of the uptake coefficient in the revised manuscript. Please see our response in **Comment #2**.

**Specific Comments #3**

*Page 9, Equation 3: Please discuss particle diameter applied.*

**Author Response:**

We have mentioned that the surface-weighted mean diameter prior to OH oxidation was used in the calculation of the uptake coefficient. Please see our response in **Comment #2**.

**Specific Comments #4**

*Page 9, Line 15–20 and Page 10, Line 1–8: It may be a too simple assumption that the ions are homogeneously distributed in small droplets. Please refer to Jungwirth and Tobias, Chem. Rev., 2006; Jungwirth et al., Chem. Phys. Lett., 2003 and subsequent studies. E.g. $SO_4^{2-}$ is likely not found at the particle surface but in the interior in contrast to the schematics in Fig. 4. Also, can it be ruled out that 3-MGA may show surfactant behavior? Even slight surfactant behavior could alter the surface concentration drastically.*

**Author Response:**

Thanks for the comment. We agree that it is too simple to assume the dissolved inorganic ions (e.g. $SO_4^{2-}$) are homogeneously distributed in the droplets with reference to the work of Jungwirth et al. (2003) and Jungwirth and Tobias (2006). Furthermore, we cannot rule out the possibility of 3-MGA being a surfactant as literatures on cloud droplet activitation indicate possible surface enhancement of dicarboxylic acid such as suberic acid in dilute aqueous droplets (Ruehl et al., 2016; Davies et al., 2019). However, literature on the surficial properties of branched dicarboxylic acid is not yet available. These factors could alter the surface concentration drastically and have not been considered. In the revised manuscript, we address that the numbers reported in this study should be considered as a first approximation to demonstrate the possible effects of AS on the surface coverage of 3-MGA. Further investigations on the surfactant properties of 3-MGA and molecular dynamic simulation are desirable to better understand the surface composition of both 3-MGA and 3-MGA–AS particles. We have added the following information in the revised manuscript to address these issues.

Page 12, Line 6, "It should acknowledge that dissolved inorganic ions (e.g. $SO_4^{2-}$) may not be homogeneously distributed in the droplets with reference to the work of Jungwirth et al. (2003) and Jungwirth and Tobias (2006). Furthermore, the surface activity of 3-MGA is not known and slight surfactant behavior could drastically alter the surface concentration. Thus the numbers presented here are to serve as a first approximation illustrating the possible effect of AS addition on the surface coverage of 3-MGA. Further investigations on the surfactant properties of 3-MGA and molecular dynamic simulation would be useful to better understand the surface composition of both 3-MGA and 3-MGA–AS particles."

Supplementary material, Page 4, Line 15, "However, it acknowledges that the assumption might not be completely correct. In accordance with the work of Jungwirth et al. (2003) and Jungwirth and Tobias (2006), the sulfate ion ($SO_4^{2-}$) likely exists in the interior of the particle instead of surface. We also cannot rule out the possibility of 3-MGA being a surfactant as literatures on cloud droplet activation indicate possible surface enhancement of dicarboxylic acid such as suberic acid in dilute aqueous droplets (Ruehl et al., 2016; Davies et al., 2019). However, literature on the surficial properties of branched dicarboxylic acid is not yet available. Further investigation on the surface activity of 3-MGA and molecular dynamic simulations are desirable to better understand the surface composition of 3-MGA and 3-MGA–AS particles."

**Specific Comments #5**

*Page 12, Line 24: Citations Petters et al. (2004) and Vereecken and Peeters (2009) are not given in bibliography and may be wrong as well?*

**Author Response:**

We are sorry for the confusion. The first citation should be Peeters et al. (2004). These two citations are chosen as references for the SAR model developed for the decomposition of alkoxy radicals. We have corrected the typo and have added the two references in the revised manuscript.

Page 15, Line 20, "Furthermore, as proposed by Peeters et al. (2004) and Vereecken and Peeters (2009), the strong hydrogen bonding among the two terminal carboxyl groups might lower the decomposition rate of the alkoxy radical."

**Specific Comments #6**

*Page 14, Line 15: ". . .over time.". Please add reference. Who has shown this?*

**Author Response:**

This is an inference from the results of this study and previous studies. We have revised the sentence and clarified that there is a possibility, yet not verified, of phase behavior change in phase-separated particles upon oxidation.

Page 17, Line 13, "Furthermore, there is a possibility that the phase separation behavior (e.g. SRH) of the particles may change in response to the change in the particle composition over time."

**Specific Comments #7**

*Page 14, Line 24: ". . .upon oxidation.". Please cite here the studies by Slade et al., ACP, 2015 and Slade et al., GRL, 2017.*

**Author Response:**

We have cited the studies mentioned in the revised manuscript.

Page 17, Line 24: "Hence, it is of interest to investigate how the phase separation characteristics of organic–inorganic particles change in response to a change in the composition upon oxidation (Slade et al., 2015; Slade et al., 2017)."

**References**

1.  Chim, M. M., Lim, C. Y., Kroll, J. H., and Chan, M. N.: Evolution in the reactivity of citric acid toward heterogeneous oxidation by gas-phase OH radicals, ACS Earth Space Chem., 2, 1323–1329, 2018.

2. Davies, J. F., Zuend, A., and Wilson, K. R.: Technical note: The role of evolving surface tension in the formation of cloud droplets, Atmos. Chem. Phys., 19, 2933–2946, 2019.

3. Jungwirth, P. and Tobias, D. J.: Specific ion effects at the air/water interface, Chem. Rev., 106, 1259–1281, 2006.

4. Jungwirth, P., Curtis, J. E., and Tobias, D. J.: Polarizability and aqueous solvation of the sulfate dianion, Chem. Phys. Lett., 367, 704–710, 2003.

5. Marsh, A., Miles, R. E. H., Rovelli, G., Cowling, A. G., Nandy, L., Dutcher, C. S., and Reid, J. P.: Influence of organic compound functionality on aerosol hygroscopicity: dicarboxylic acids, alkyl-substituents, sugars, and amino acids, Atmos. Chem. Phys., 17, 5583–5599, 2017.

6. Peeters, J., Fantechi, G., and Vereecken, L.: A generalized structure-activity relationship for the decomposition of (substituted) alkoxy radicals, J. Atmos. Chem., 48, 59–80, 2004.

7. Ruehl, C. R., Davies, J. F., and Wilson, K. R.: A interfacial mechanism for cloud droplet formation on organic aerosols, Science, 351, 1447–1450, 2016.

8. Slade, J. H., Thalman, R., Wang, J., and Knopf, D. A.: Chemical aging of single and multi-component biomass burning aerosol surrogate particles by OH: implications for cloud condensation nucleus activity, Atmos. Chem. Phys., 15, 10183–10201, 2015.

9. Slade, J. H., Shiraiwa, M., Arangio, A., Su, H., Pöschl, U., Wang, J., and Knopf, D. A.: Cloud droplet activation through oxidation of organic aerosol influenced by temperature and particle physical state, Geophys. Res. Lett., 44, 1583–1591, 2017.

10. Worsnop, D. R., Morris, J. W., Shi, Q., Davidovits, P., and Kolb, C. E.: A chemical kinetic model for reactive transformations of aerosol particles, Geophys. Res. Lett., 29(20), 1996, doi:10.1029/2002GL015542, 2002.

---

## Author Comment (AC2) · 29 Jun 2019

*This study reports differences in the heterogeneous OH oxidation kinetics and particle-phase products of representative pure-component organic aerosol and mixed organic-inorganic aerosol under conditions of particle deliquescence. The authors report the reactive uptake coefficients between OH + methylglutaric acid and OH + methylglutaric acid/ammonium sulfate aerosol measured using an oxidation flow cell coupled with a direct analysis in real time (DART) mass spectrometer and conclude that while oxidation products are similar between the two aerosol systems, the uptake kinetics are significantly slower in the case of the methylglutaric acid/ammonium sulfate aerosol mixture. Overall, the manuscript is well written, the topic is of interest, and the study appears to be sound/not overstated. I recommend publication in ACP after the authors respond to the following comments.*

**We would like to sincerely thank the reviewer for his/her thoughtful comments and suggestions. Please see our responses to reviewer's comments and suggestions below.**

**Comment #1**

*Page 2, line 4: Be careful not to understate the role of dissolved salts – this study was performed on deliquesced particles at 85% relative humidity. Because of the hygroscopicity of AS, 3-MGA–AS will deliquesce at a lower relative humidity than pure 3-MGA particles. This may not significantly affect the reaction mechanism itself, but diffusion of reactants from the particle bulk to the surface may be quite different at a different relative humidity, thus the extent of reaction I would expect depends also on the diffusivity of the reactants, which may be more important under more relevant daytime relative humidity (when [OH] peaks in the real environment). Written like this suggests the inorganic component has no influence on the reaction.*

**Author Response:**

Thanks for the comment. We did experiments to show that the water activity of saturated 3-MGA solution is 0.935 (i.e. DRH = 93.5%). The effloresced 3-MGA–AS particles absorbed water gradually upon moistening, with the AS rich phase dissolved at ≤ 80% RH and the remaining fully deliquesced at ~85 % RH. These observations agree with the reviewer's comment that 3-MGA–AS particles deliquesce at a lower RH than pure 3-MGA particles. We acknowledge that this work only investigates the role of dissolved salts in the heterogeneous reactivity of well-mixed aqueous organic–inorganic droplets at a sufficiently high relative humidity (85%). The effects of inorganic salts on the heterogeneous reactivity could vary greatly, depending on the particle composition and environmental conditions (e.g. RH and temperature). For instance, as pointed out by the reviewer, the diffusivity of species from bulk to the surface plays an important role in the extent of heterogeneous reaction. For aqueous 3-MGA–AS particles at 85 % RH in our work, we are not certain about the effect of AS on the particle

viscosity, but the viscosity is expected to be in the liquid-like regime due to the substantial water content. We postulate that the particle viscosity may decrease due to the increase in water uptake by the addition of hygroscopic, dissolved AS, which allows for faster bulk diffusion of species. Therefore, 3-MGA–AS particles are likely well-mixed during oxidation and no substantial bulk diffusion limitation is expected in our case. On the other hand, at lower RH, aqueous 3-MGA–AS droplets likely become more concentrated and more viscous before efflorescence, possibly giving rise to diffusion limitation during oxidation. To our best knowledge, the effect of inorganic salts on the particle viscosity of organic–inorganic mixed phases remains largely unexplored. Further studies on the effect of inorganic salts on diffusivity of the reactants within organic–inorganic particles are warranted. We have revised the manuscript and added the information above.

Page 2, Line 13, "Our results suggest that inorganic salts likely alter the overall heterogeneous reactivity of organic compounds with gas-phase OH radicals rather than reaction mechanisms in well-mixed aqueous organic–inorganic droplets at a high humidity (i.e. 85% RH). It also acknowledges that the effects of inorganic salts on the heterogeneous reactivity could vary greatly, depending on the particle composition and environmental conditions (e.g. RH and temperature). For instance, at lower relative humidities, aqueous 3-MGA–AS droplets likely become more concentrated and more viscous before efflorescence, possibly giving rise to diffusion limitation during oxidation under relatively dry or cold conditions. Further studies on the effects of inorganic salts on the diffusivity of the species under different relative humidities within the organic–inorganic particles are also desirable to better understand the role of inorganic salts in the heterogeneous reactivity of organic compounds."

**Comment #2**

*Page 5, lines 10-12: Presumably, quantification of [OH] was done prior to addition of the aerosol particles to the flow cell? The reported second-order rate coefficients for heterogeneous OH oxidation of 3-MGA and 3-MGA–AS ($2.72$-$3.26\times10^{-12}$ $cm^3$ $molecule^{-1}$ $s^{-1}$) are competitive with that for hexane ($5.21\times10^{-12}$ $cm^3$ $molecule^{-1}$ $s^{-1}$) (Atkinson, 2003). Please specify. If mixed, how would this affect the determination of [OH]?*

**Author Response:**

In this study, the OH exposure was determined by the in-situ measurement of the decay of gas-phase hexane in the presence of particles inside the reactor using a gas chromatograph coupled with a flame ionization detector (GC-FID). Hence, the impacts of competitions between the heterogeneous oxidation and the gas-phase oxidation on the quantification of OH concentration have been taken into account. We have revised the manuscript to clarify the point.

Page 6, Line 8, "Since the OH exposure (and OH concentration) was determined by the in situ measurement of the decay of gas-phase hexane inside the reactor, the impacts of RH and water uptake by particles inside the reactor on the generation and concentration of gas-phase OH radicals have been taken into account. The competitions between the heterogeneous oxidation and the gas-phase oxidation have also been considered when quantifying OH concentration."

**Comment #3**

*Page 5, lines 21-23: Please indicate where the relative humidity measurement was made in the setup. Do you expect the particles that get sampled through the inlet of the DART instrument to be at a different relative humidity than when they were oxidized? From personal experience, Carulite catalyst can decrease the relative humidity. Please comment on potential variations in the relative humidity as part of the experimental setup and whether it has an effect on the products analyzed.*

**Author Response:**

The relative humidity (RH) was measured at the inlet of the reactor. We have not measured the RH of the particle stream before and after the Carulite catalyst denuder and do not know to what extent the humidity of the particle stream dropped. We agree with the reviewer that the RH inside the reactor was slightly higher than RH after passing through the Carulite catalyst denuder. Although the particles got sampled at a different RH during the DART analysis, this would not have significant effect on the reaction products analyzed because the decrease in RH after oxidation would not significantly affect the formation of reaction products which primarily occurred inside the reactor. We have revised the manuscript and added the information above.

Page 5, Line 14, "The RH within the reactor was controlled by varying the dry $N_2$ to humidified $N_2$ ratio and was measured at the inlet of the reactor."

Page 6, Line 15, "It acknowledges that Carulite catalyst denuder can slightly decrease the RH of the particle stream. However, this would not have significant effect on the reaction products analyzed, because the decrease in RH after oxidation would not significantly affect the formation of reaction products, which primarily occurred inside the reactor."

**Comment #4**

*Page 6, lines 1-4: It is known that thermal desorption methods lead to inaccurate estimates of particle volatility, e.g., Stark et al. (2017) demonstrate that many organic acids and alcohols, common constituents of secondary organic aerosol, can decompose at temperatures as low as 200 ∘C. In that study, a significant fraction of detected compounds resulted from thermal decomposition, suggesting the detected fragments were not actually present in the atmosphere, but rather formed during decomposition in the inlet of the instrument. Please discuss potential caveats of the thermal desorption technique used here and its impact on the observed product distribution.*

**Author Response:**

Thanks for the comments. The thermal composition of particle-phase products could possibly occur during the thermal desorption processes in our chemical analysis. The mass spectra of some organic acids and alcohols (e.g. succinic acid, ketosuccinic acid and tartaric acid) are available in the work of Chan et al. (2014), showing insignificant thermal decomposition during the DART analysis. In this study, the thermal decomposition of 3-MGA was found to be insignificant before oxidation as the deprotonated molecular ion of 3-MGA is the dominant peak in the mass spectra (Figure 1). We acknowledge that the reactions between peroxy radicals may yield organic peroxides and oligomers, which may decompose thermally. We cannot completely rule out the possibility of that, but there was no indication of any fragment ions expected from the thermal decomposition in the mass spectra. Together, these results suggest that the impact of thermal decomposition on the observed product distribution is likely insignificant. This information is added in the revised manuscript

Page 7, Line 17, "It acknowledges that thermal composition of reaction products could possibly occur during the thermal desorption process (Stark et al. 2017). The mass spectra of some organic acids and alcohols (e.g. succinic acid, ketosuccinic acid and tartaric acid) are available in the work of Chan et al. (2014), showing insignificant thermal decomposition during the DART analysis. In this study, the thermal decomposition of 3-MGA was found to be insignificant as the deprotonated molecular ion of 3-MGA is the dominant peak before oxidation in the mass spectra (**Figure 1**). We acknowledge that reactions between peroxy radicals may yield organic peroxides and oligomers, which may decompose thermally. We cannot completely rule out the possibility of such reactions, but there was no indication of any fragment ions expected from the thermal decomposition in the mass spectra. Together, these results suggest that the impact of thermal decomposition on the observed product distribution is likely insignificant."

**Comment #5**

*Page 9, lines 4-5: Please specify whether these studies were performed using mono or polydisperse aerosol. Are the reported diameters a median value or geometric mean? Also, are the reported diameters number- or surface area-weighted? Please comment on the effect of particle size, e.g., regarding evaporation (Vaden et al., 2011) and mixing timescales of volatile/semi-volatile components (Meng and Seinfeld, 1996).*

**Author Response:**

Thanks for the comment. Polydisperse particles were used in this work. The diameter reported is the surface area-weighted mean diameter of these particles. Upon oxidation, fragmentation products evaporate and partition back to the gas phase. Vaden et al. (2011) have discussed that evaporation of highly viscous particles is likely independent of particle size distribution and is unlikely to be significantly influence the overall evaporation behavior. As the study of Vaden et al. (2011) focused on highly viscous particles while the focus of this study is more liquid-like particles, their results may not be applicable in our study. Since 3-MGA-AS particles are more liquid-like particles, the evaporate rate would scale with the total surface area of the polydisperse particle population. Since the spread of the polydisperse particle population is small in this work, the size change is not likely substantial with regard to determining $\gamma_{eff}$ as the total particle surface area did not change dramatically. In the work of Meng and Seinfeld (1996), the mixing timescales of volatile/semi-volatile species are evaluated. Although it was suggested by the study that the timescales may increase with increasing particle size, the difference may not be that significant in our study as the span of the polydisperse particles is much smaller than the difference between coarse particles and fine particles used in their studies. We thus postulate that the spread of particle size and the mixing timescale would not play a role in the evaporation of fragmentation products during oxidation. We have revised the manuscript.

Page 10, Line 16, "The surface-weighted mean diameters prior to OH oxidation (203.0 nm for 3-MGA and 200.8 nm for 3-MGA–AS, respectively) are used in the calculation of $\gamma_{eff}$. Upon oxidation, the surface-weighted mean diameters decreases from 203.0 nm to 170.7 nm for 3-MGA particles and decreases from 200.8 nm to 187.8 nm for 3-MGA–AS particles (**Figure S5**, *supplementary material*). The decrease in the particle diameter upon oxidation is likely attributed to the formation and volatilization of fragmentation products and the associated evaporative loss of water molecules. Vaden et al. (2011) have discussed that evaporation of highly viscous particles is likely independent of particle size distribution and is unlikely to be significantly influence the overall evaporation behavior. As the study of Vaden et al. (2011) focused on highly viscous particles while the focus of this study is more liquid-like particles, their results may not be applicable in our study. Since 3-MGA-AS particles are more liquid-like particles, the evaporate rate would scale with the total surface area of the polydisperse particle population. Since the spread of the polydisperse particle population is small in this work, the size change is not likely substantial with regard to determining $\gamma_{eff}$ as the total particle surface area did not

change dramatically. In the work of Meng and Seinfeld (1996), the mixing timescales of volatile/semi-volatile species are evaluated. Although it was suggested by the study that the timescales may increase with increasing particle size, the difference may not be that significant in our study as the span of the polydisperse particles is much smaller than the difference between coarse particles and fine particles used in their studies. We thus postulate that the spread of particle size and the mixing timescale would not play a role in the evaporation of fragmentation products during oxidation."

**Comment #6**

*Page 9, lines 11-12: Isn't γeff for 3-MGA–AS more than twice as small as that for 3- MGA, rather than 59%? The relative percent difference is 59%. This is a bit confusing as written. Please consider rephrasing. Did the authors measure the aerosol size distribution after OH oxidation? If so, is there evidence of particle mass growth (functionalization) or volatilization (fragmentation)? This can be assessed simply by plotting the ratio of initial aerosol volume to final aerosol volume as a function of OH exposure. Such an analysis would be a valuable addition to the paper. The authors provide a reasonable argument for the difference in γeff between 3-MGA and 3-MGA–AS particles, but I would caution extension (i.e., overall applicability) of Eq. S1 to other aerosol systems at different relative humidity. Equation S1 is an oversimplification of the likely complex interactions and concentration gradients present in atmospheric aerosol. In this study, Eq. 1 simply indicates there is less 3-MGA in the 3-MGA–AS mixture to react with OH compared to the pure 3-MGA particles. However, if the particles were phase-separated or exhibited core-shell structure, e.g., at low relative humidity, does Eq. S1 indicate what is at the surface?*

**Author Response:**

Thanks for the comments. We have rephrased the sentence in the revised manuscript that the values of $\gamma_{eff}$ for 3-MGA particles is larger than that of 3-MGA–AS particles by about 2.4 times.

The size distribution was measured after OH oxidation using the SMPS. The size decreases upon oxidation from 203 to 171 nm for 3-MGA and from 201 to 188 nm for 3-MGA–AS. This may be an indicator of the possible volatilization (fragmentation) processes. We have added the plot of particle size against OH exposure (**Figure S5**, *supplementary material*) in the supplementary material.

[Figure]

**Figure S5**. The change in surface-weighted mean diameter as a function of OH exposure for 3-MGA particles and 3-MGA–AS particles, respectively.

We agree with the reviewer that Eqn. S1 may not be applicable to providing explanations for particles with complex interactions and concentration gradients such as phase separation. This

equation would fail for predicting the surface composition if the particles were phase-separated or exhibited core-shell structure. Understanding the surface activity of 3-MGA and performing molecular dynamic simulations are desirable to better understand the surface composition of the particles with different phases (e.g. well-mixed homogeneous liquid phase or phase separated). We have added the following information in the revised manuscript.

Page 6, Line 18, "Size distribution of the particles was determined by sampling a small portion of the particle stream using a scanning mobility particle sizer (SMPS, TSI) after oxidation took place."

Page 10, Line 16, "The surface-weighted mean diameters prior to OH oxidation (203.0 nm for 3-MGA and 200.8 nm for 3-MGA–AS, respectively) are used in the calculation of $\gamma_{eff}$. Upon oxidation, the surface-weighted mean diameters decreases from 203.0 nm to 170.7 nm for 3-MGA particles and decreases from 200.8 nm to 187.8 nm for 3-MGA–AS particles (**Figure S5**, *supplementary material*). The decrease in the particle diameter upon oxidation is likely attributed to the formation and volatilization of fragmentation products and the associated evaporative loss of water molecules."

Page 11, Line 23, "The value of $\gamma_{eff}$ for 3-MGA particles is larger than that of 3-MGA–AS particles by about 2.4 times."

Supplementary material, Page 4, Line 15, "However, it acknowledges that the assumption might not be completely correct. In accordance with the work of Jungwirth et al. (2003) and Jungwirth and Tobias (2006), the sulfate ion ($SO_4^{2-}$) likely exists in the interior of the particle instead of surface. We also cannot rule out the possibility of 3-MGA being a surfactant as literatures on cloud droplet activation indicate possible surface enhancement of dicarboxylic acid such as suberic acid in dilute aqueous droplets (Ruehl et al., 2016; Davies et al., 2019). However, literature on the surficial properties of branched dicarboxylic acid is not yet available. Further investigation on the surface activity of 3-MGA and molecular dynamic simulations are desirable to better understand the surface composition of 3-MGA and 3-MGA–AS particles. It is also noted that Eqn. S1 may not be applicable to providing explanations for particles with complex interactions and concentration gradients such as phase separation. This equation would fail for predicting the surface composition if the particles were phase-separated or exhibited core-shell structure."

*Supporting material*, we have added the **Figure S5** in the *supporting material* to illustrate the change in particle diameter for 3-MGA particles and 3-MGA–AS particles upon oxidation

**References**

1. Davies, J. F., Zuend, A., and Wilson, K. R.: Technical note: The role of evolving surface tension in the formation of cloud droplets, Atmos. Chem. Phys., 19, 2933–2946, 2019.
2. Meng, Z. Y., and Seinfeld, J. H.: Time scales to achieve atmospheric gas-aerosol equilibrium for volatile species, Atmos. Environ., 30, 2889-2900, 1996.
3. Ruehl, C. R., Davies, J. F., and Wilson, K. R.: A interfacial mechanism for cloud droplet formation on organic aerosols, Science, 351, 1447–1450, 2016.
4. Stark, H., Yatavelli, R. L. N., Thompson, S. L., Kang, H., Krechmer, J. E., Kimmel, J. R.,Palm, B. B., Hu, W. W., Hayes, P. L., Day, D. A., Campuzano-Jost, P., Canagaratna, M. R., Jayne, J. T., Worsnop, D. R., and Jimenez, J. L.: Impact of Thermal Decomposition on Thermal Desorption Instruments: Advantage of Thermogram Analysis for Quantifying Volatility Distributions of Organic Species, Environ. Sci. Technol., 51, 8491-8500, 2017.

5. Vaden, T. D., Imre, D., Beranek, J., Shrivastava, M., and Zelenyuk, A.: Evaporation kinetics and phase of laboratory and ambient secondary organic aerosol, Proceedings of the National Academy of Sciences of the United States of America, 108, 2190-2195, 10.1073/pnas.1013391108, 2011.